# Structure of the TnsB transposase-DNA complex of type V-K CRISPR-associated transposon

Francisco Tenjo-Castaño[1], Nicholas Sofos[1], Blanca López-Méndez [2], Luisa S. Stutzke[1], Anders Fuglsang [1], Stefano Stella [1,3] & Guillermo Montoya [1] ✉

CRISPR-associated transposons (CASTs) are mobile genetic elements that co-opted CRISPR-Cas systems for RNA-guided transposition. Here we present the 2.4 Å cryo-EM structure of the *Scytonema hofmannii* (sh) TnsB transposase from Type V-K CAST, bound to the strand transfer DNA. The strand transfer complex displays an intertwined pseudo-symmetrical architecture. Two protomers involved in strand transfer display a catalytically competent active site composed by DDE residues, while other two, which play a key structural role, show active sites where the catalytic residues are not properly positioned for phosphodiester hydrolysis. Transposon end recognition is accomplished by the NTD1/2 helical domains. A singular in trans association of NTD1 domains of the catalytically competent subunits with the inactive DDE domains reinforces the assembly. Collectively, the structural features suggest that catalysis is coupled to protein-DNA assembly to secure proper DNA integration. DNA binding residue mutants reveal that lack of specificity decreases activity, but it could increase transposition in some cases. Our structure sheds light on the strand transfer reaction of DDE transposases and offers new insights into CAST transposition.

The discovery of an adaptive prokaryotic immune system called Clustered Regularly Interspaced Short Palindromic Repeats (CRISPR), in which the repeats associate with Cas (CRISPR-associated) proteins, has constituted a revolution in life sciences. CRISPR-Cas systems are highly diverse ribonucleoprotein (RNP) complexes with different evolutionary origins. They are divided into two classes, Class 1 and Class 2, the former including a multi-subunit effector complex and the later a single protein effector[1]. The two classes are further divided into six types (I–VI) depending on the identity of the nuclease module, and many subtypes depending on which other Cas proteins are present in other functional modules. Especially, Class 2 members have garnered much attention as they have been

developed into versatile RNA-guided nucleases for RNA-guided genome editing, which has radically altered life sciences, enabling genome manipulation in living organisms[2].

Recently, several CRISPR-Cas machineries have been found associated with Tn7-like transposon systems in types I, IV and V. These CAST systems[3,4] are a product of an evolutionary process by which Tn7-like transposons recruited the CRISPR-Cas system for transposon mobilization. These complexes do not degrade their target DNA and operate exclusively in prokaryotes. They insert large DNA cargos (10–30 kb) at specific genome regions without the need for homology-directed repair[4–7], combining the site-selection precision of CRISPR-Cas with the integration properties of transposons[8]. Therefore, CASTs

[1]Structural Molecular Biology Group, Novo Nordisk Foundation Centre for Protein Research, Faculty of Health and Medical Sciences University of Copenhagen, 2200 Copenhagen, Denmark. [2]Protein Purification and Characterisation Facility, Novo Nordisk Foundation Centre for Protein Research, Faculty of Health and Medical Sciences University of Copenhagen, 2200 Copenhagen, Denmark. [3]Present address: Twelve Bio ApS, Ole Maaløes Vej 3, 2200 Copenhagen, Denmark. ✉e-mail: guillermo.montoya@cpr.ku.dk

are thought to provide a very promising system for the development of next-generation gene-editing tools.

CAST I-F, I-B and V-K subtypes, from *Vibrio cholerae* (vc), *Anabaena variabilis* (av) and *Scytonema hofmannii* (sh), respectively, were the first ones to be discovered[3,4], but recent bioinformatic searches of metagenomic databases have vastly expanded the known CAST *repertoire* to over 1000 non-redundant subsystems representing Types I, IV and V[6]. To date, all known CASTs are derived from Tn7-like transposons and include the corresponding crRNAs and *Cas* genes necessary for target selection[6,7], and the core transposition machinery within a Tn7-like transposon locus. This includes TnsB, TnsC, TniQ (a homologue of *E. coli* TnsD), and, in certain cases, TnsA genes. In an analogy with the Tn7 transposon systems, the CAST Tn7 proteins are thought to assemble into a pre-integration nucleoprotein complex that involves TnsA (in Types I and IV), TnsB, TnsC and TniQ, to regulate transposition into the insertion site. TnsA is an endonuclease that cleaves the 5′-ends of the transposon[9] and interacts with TnsB, TnsC and DNA[9–11]. TnsB is a recombinase and catalyses the cleavage of the 3′-ends of the transposon. In the canonical Tn7 system, the interaction of TnsA and TnsB is necessary to activate catalysis[12]. TnsC is part of the AAA+ ATPase family and directs TnsA/TnsB to the insertion site[11,13]. The interaction of TniQ with the target DNA bound by the CRISPR-Cas complex is thought to create a distortion in the DNA structure, allowing TnsC to recognize both TniQ and the DNA[14], and leading to the insertion of the transposon at the attachment site. However, type V-K CAST systems are different from Tn7 due to the lack of TnsA in their loci. Since type V-K CAST does not comprise any other protein to replace TnsA endonuclease activity, its transposition produces cointegrates that need to be resolved[15,16]. Cointegrate resolution is presumably carried out by RecA-dependent recombination, as is the case for other transposons lacking an enzyme with a 5′ endonuclease activity. Nonetheless, transposons from the Tn5053 family, which also lack TnsA while comprising homologs for TnsB, TnsC and TniQ, can resolve co-integrates using a transposon-encoded resolvase TniR. Whether a host-encoded resolvase could supply this activity in type V-K CAST is yet to be determined.

TnsB transposases belong to the retroviral integrase superfamily with an RNase H fold catalytic domain and a DDE active site motif. TnsB associates with the left and right ends of the transposon and catalyses their cleavage generating free 3′-hydroxyl groups, which are later used in a nucleophilic attack downstream of the target sequence[17]. Finally, it performs the strand-transfer reaction that leads to the insertion of the DNA cargo into the target site.

The shTnsB transposase shares homology with other members of the DDE integrase family such as *E. coli* TnsB, MuA and Tn5053 (Supplementary Fig. 1). The structure of *E. coli* (ec) TnsB in complex with the end of the transposon has provided new evidence explaining differences in recognition of the left and right ends of the element[18]. However, the sequence identity of ecTnsB with shTnsB is low, thus making it difficult to understand the details of the integration mechanism of CASTs and other transposon systems containing integrases of the DDE family.

To comprehend how shTnsB facilitates RNA-guided transposition in the CAST V-K system, we determine the structure of the shTnsB-DNA complex captured after the strand-transfer reaction by single-particle cryo-EM at 2.46 Å resolution (Fig. 1). The structure reveals an entangled architecture of the shTnsB protein around the DNA building a pseudo-symmetrical assembly in which the four subunits of shTnsB can be grouped in two different conformations. The DNA in the DDE catalytic pockets displays a sharp bent after the strand-transfer reaction. Site-directed mutagenesis and in vivo transposition assays reveal the important role in the transposition of key DNA-binding residues. The strand-transfer reaction complex of shTnsB opens new avenues to understanding RNA-guided transposition in CAST systems.

## Results

### Isolation of shTnsB and generation of the strand-transfer complex (STC)

The recombinant shTnsB protein was expressed in *E. coli* and purified using a combination of affinity and size exclusion chromatography (SEC). The protein behaved as a monomer in SEC-MALS displaying a MW of 68 kDa (Supplementary Fig. 2a, b, Methods). We analysed its DNA-binding properties by EMSA using oligonucleotides with a different number of short and long terminal repeats (SR and LTR, respectively) present on the left and right ends (LE and RE) of the transposon sequences (Supplementary Fig. 2c, d, Supplementary Table 1). The analysis of the band shifts with all the different substrates revealed a ladder of discrete bands, which showed relative mobility inversely dependent on the concentration of the protein, suggesting the binding of one protein monomer per repeat. Assemblies higher than six or seven proteins could not be observed as their size prohibited entry into the acrylamide gel (Supplementary Fig. 2c). Then, we tested whether shTnsB could bind RE or LE independently. Both complexes, shTnsB-RE and shTnsB-LE, were detected, indicating that the presence of both ends is not required for shTnsB binding to the repeats (Supplementary Fig. 2d). Similarly, the number of bands observed when shTnsB was mixed with RE corresponds to 5 TnsB binding sites. However, the number of bands detected when shTnsB was mixed with LE was 4, i.e. one higher than the expected three binding sites. This could be explained by the interaction of two shTnsB:LE complexes similar to the one observed in the STC complex (Fig. 1), or in other transposase structures not bound to the target DNA, such as in Tn5[19]. However, aggregation of the shTnsB:LE complex could also cause a band shift and cannot be excluded as a possibility. Finally, we tested the binding of shTnsB to dsDNA containing only LTR(6)-SR(1) (Supplementary Fig. 2e) and observed two bands corresponding to two shTnsB binding sites as expected. Furthermore, we incubated this complex in a buffer containing different NaCl concentrations at 37 and 45 °C to determine whether these variables might change shTnsB affinity. However, the association was not affected by these changes. Altogether, the EMSA results suggested that shTnsB binds every repeat present in both RE and LE in a cargo-independent way.

Initially, we prepared cryo-EM grids and collected data using a sample containing shTnsB and both RE and LE (i.e. two dsDNA sequences, one with the RE sequence and one with the LE sequence with no cargo, Supplementary Fig. 2e). This sample was heterogeneous and suffered from preferential orientation. The processing of these data rendered low-resolution reconstructions of insufficient quality to build an atomic model (Supplementary Fig. 3). However, two protuberances could be observed associated with an elongated density, suggesting the presence of two shTnsB protomers bound to a dsDNA face of a SR-LTR set. The elongated density assigned to the DNA is bent in a similar manner as in the recently published structure of the RE bound ecTnsB[18], further supporting that the low-resolution map corresponds to a pre-transposition complex.

To stabilize a shTnsB-DNA complex, we designed oligonucleotides to reconstitute the STC, i.e. the state representing posttransposition instead of pre-transposition as described above (Methods, Supplementary Table 1 and Supplementary Fig. 4). The target sequences for the reconstituted complex were chosen based on the natural sequences flanking the attachment site in the genome of *S. hofmannii* UTEX 2349. This strategy is similar to the one followed to obtain the P element transpososome structure[20] except that it does not use a symmetrical STC DNA but the natural STC asymmetric sequences of *S. hofmannii* UTEX 2349, to represent the native complex. The STC DNA comprises two transferred strands (TS) and non-transferred strands (NTS) representing LTR(8)-SR(5) and SR(1)-LTR(6) dsDNA bound to the attachment site of CAST (Supplementary Fig. 4). A distance of 5 bp between the

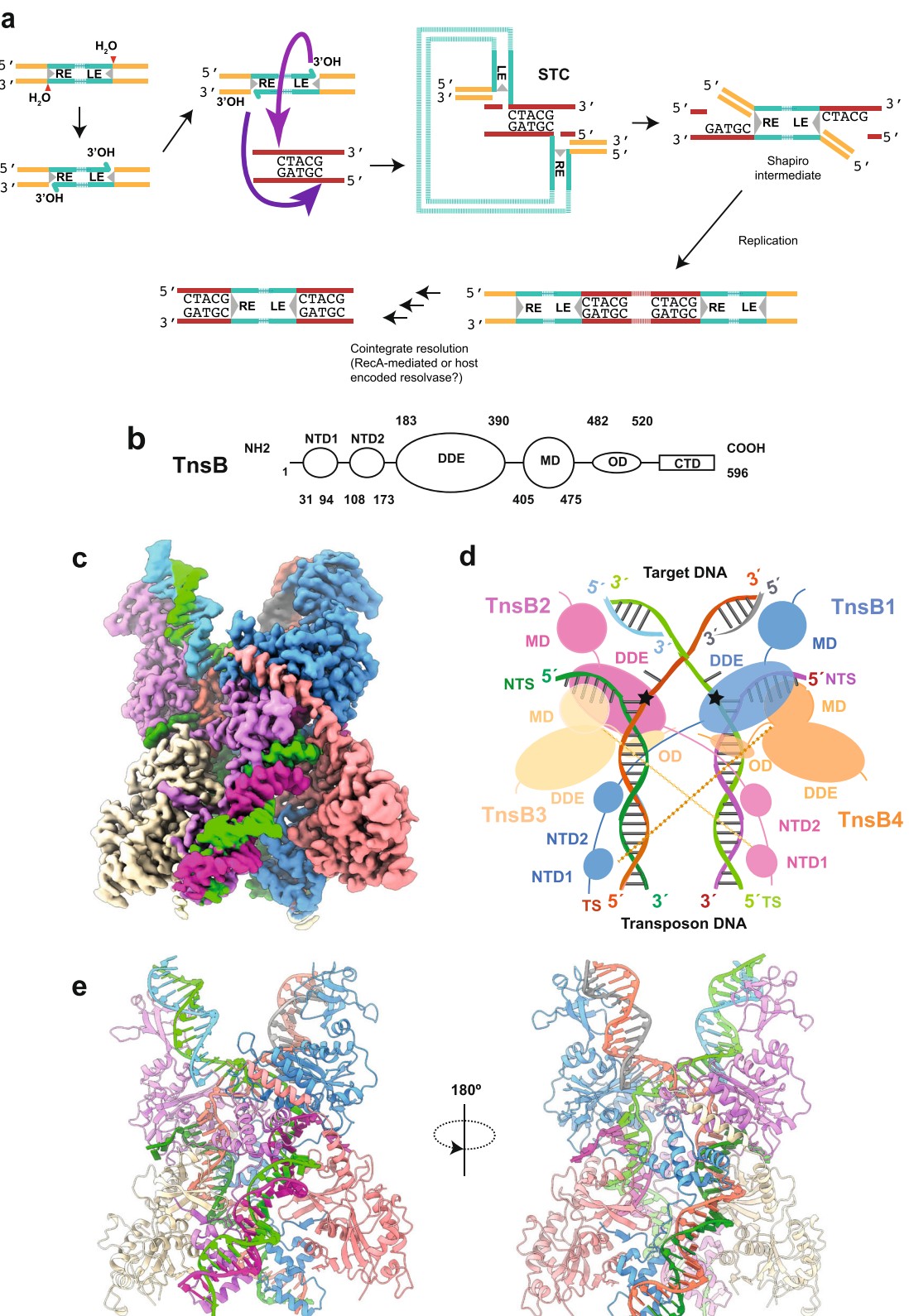

insertion sites of LTR(8)-SR(5) and SR(1)-LTR(6) was chosen as CAST transposition causes a 5 bp duplication in the insertion site[21] (Fig. 1a).

The reconstituted DNA STC was mixed with the purified protein, and the assembly of the shTnsB-STC complex was verified by SEC-MALS (Supplementary Fig. 2b). Two peaks were observed, the lower MW peak contained the unbound protein and DNA used in the

reconstitution, while the high MW peak, which eluted at 358.7 kDa, coincided very well with the association of the STC DNA (104.6 kDa) and 4 protomers of shTnsB (68 kDa), whose theoretical MW is 376.6 kDa. This assembly was subjected to single-particle analysis by cryo-EM, which rendered a 2.46 Å resolution map where we built an atomic model of the shTnsB-STC complex in its post-catalytic state (Fig. 1a).

**Fig. 1 | Cryo-EM structure of the CAST shTnsB subunit representing the post-catalytic state of the strand-transfer complex. a** General scheme of the integration reaction followed by the TnsB family of transposases. The DDE active site catalyses the attack of $H_2O$ at the transposon ends. The direct attack of the 3´OH of the DNA cargo on the target DNA site produces the strand-transfer complex (STC). The Shapiro intermediate is resolved by replication producing a target-site duplication and a cointegrate[25], which can be resolved via RecA or a resolvase. **b** Architecture of the shTnsB protein. **c** Cryo-EM density map at 2.46 Å global resolution of the TnsB (STC) in a post-catalytic state (See also Supplementary

Figs. 6–7, Supplementary Table 2). The map is coloured according to the cartoon describing the structure in panel **d. d** Cartoon of the shTnsB-STC. The complex is composed by 4 shTnsB protomers and 6 DNA oligonucleotides representing the post-catalytic state of the STC (panel **a**). In this view the target DNA is located at the top of the structure while the longer transposon ends are located at the bottom. The non-transferred strand (NTS) and the transferred strand (TS) of the transposon are labelled in the scheme. The dashed lines depict the interaction between the MDs and NTD1s due to the curvature of the DNA branches. **e** Ribbon diagram displaying an overview of the shTnsB-STC assembly.

## The overall structure of shTnsB-STC complex

The shTnsB polypeptide is composed of an N-terminal domain that can be divided into two subdomains (NTD1 and NTD2), a catalytic DDE domain (DDE), a middle domain (MD), an oligomerisation domain (OD) and a C-terminal domain (CTD) (Fig. 1b). The shTnsB architecture resembles that of Mu transposase (MuA)[22], even though sequence identity is limited to the catalytic DDE domain (Supplementary Fig. 1a). We collected 4728 movies and set coordinates for 9.6 million particles, which were reduced to 415k after 2D classification. Using this set of particles, we performed non-uniform refinement, 3D variability analysis and subsequently heterogeneous refinement in cryoSPARC[23]. This approach yielded two maps at 2.5 and 2.8 Å. Further rounds of refinement and 3D variability analysis using the 260k particles of the 2.5 Å map rendered a cryo-EM map at 2.46 Å global resolution that allowed the modelling of the shTnsB-STC complex (Fig. 1c, Supplementary Fig. 5–7, Supplementary Table 2 and Methods). However, the large flexibility of the DNA compromised the visualization of the DNA ends containing SR(5) and SR(1), preventing the determination of possible contacts with shTnsB (Supplementary Fig. 7, Supplementary Movie 1 and Supplementary Table 1).

The structure of the shTnsB-STC complex resembles an elongated X with curved arms of different lengths. The longer arms correspond to the transposon ends, while the short ones belong to the bent target DNA (Figs. 1d, e, 2a). The four protomers of shTnsB (shTnsB1–4) display an intertwined assembly on the STC DNA. The protein is found in two different conformations to associate with the STC DNA and build the complex (Fig. 2b). shTnsB1 and 2 depict an extended conformation along the branched nucleic acid structure, where we could not observe the OD and CTD domains (Fig. 1b, d). In both protomers, each of the NTD1 and NTD2 domains is associated with LTR(8) and LTR(6), while the catalytic DDE domain is visualized in trans at the junction between the target DNA and the transposon ends. This positioning in trans of the catalytic domain is common in DNA transposition, as it makes phosphodiester cleavage dependent on complex assembly (Figs. 1e, 2b). On the target DNA side, the assembly is stabilized by interactions of the backbone with the MD and DDE domains of the shTnsB1 and 2 protomers (R416, Q427, K290, N428) (Fig. 2b). Thus, shTnsB1 accomplishes DNA recognition in the LTR(8) end, and its DDE domain catalyses the attack of the 3´OH in the LTR(6) branch on the target DNA, promoting the strand-transfer reaction. A symmetric arrangement of the STC DNA is observed for shTnsB2, which catalyses the strand transfer of the LTR(8) on the other strand of the target DNA (Figs. 1a, 2b).

The shTnsB3 and 4 protomers are arranged in a different conformation with few contacts with the backbone of the DNA (Fig. 2b). The NTD1 and NTD2 domains of these protomers are not observed in our structure, suggesting that they are highly mobile. In addition, the DDE domain is pulled out of the DNA while the MD accommodates the NTS regions from the LTR(6) and LTR(8) ends after catalysis. The structure revealed the important role of these two protomers in the association of shTnsB1 and shTnsB2 with the DNA, as in this conformation, they show an ordered OD domain. The helix between S502 and S522 is not observed in shTnsB1-2, but in the conformation adopted by shTnsB3-4, the helix facilitates the assembly of the STC complex by intertwining the NTD2 of

shTnsB1 and shTnsB2, with the DDE domain of the opposite protomer in each branch of the DNA, thereby, linking the recognition of the two transposon ends with the catalytic sites of shTnsB1-2 (Figs. 1d, 2b).

## shTnsB-DNA recognition

The recognition of LTR(6) and LTR(8) is performed by the NTD1 and 2 helical domains of the shTnsB1 and 2 subunits (Fig. 2c). The NTD1 domain structure reveals a helix-turn-helix Myb/homeodomain-like fold. This domain is a member of the homeobox superfamily of transcription factors[24]. Direct base contacts by the major groove binding helix principally account for the sequence-specific recognition. The R77, which is well-conserved in cyanobacterial homologues and in Tn5053 but substituted by lysine in MuA and alanine in ecTnsB (Supplementary Fig. 1), makes polar contacts with dG37, while R81 associates with the bases of the consecutive dG36 and dT37 nucleotides on the complementary strand of the LTR(6) in the LE (Supplementary Fig. 7, 8 and Supplementary Table 1). Other residues interacting with the DNA (R58, R66, K84 and T78) show polar interactions with the backbone of both DNA strands.

The NTD1 is joined to NTD2 by a long loop that runs along the groove of the DNA. The NTD2 domain is functionally similar to the second DNA-binding domain in other transposases. However, it displays limited conservation to the closest homologues of shTnsB (Supplementary Fig. 1), and its structure resembles the paired domain found in Paired box (Pax) genes. The well-conserved R99 in that loop builds polar interactions with the bases of dT31 and dA32, and dA45, while R106 also recognizes the bases of the dT30 and dT46 in different strands. The rest of the associations of the loop with the DNA involve backbone contacts. The NTD2 domain also binds to the DNA in the major groove. However, it displays fewer specific contacts with the bases. Only R158 and K154 associate with dA48-dG49 and dG24, respectively, while the rest of the residues associated with the backbone of the nucleic acid. These interactions are similar in the NTD1 and 2 domains on the LTR(6) and LTR(8) branches of the protein-DNA complex (Supplementary Fig. 7).

## Assembly of the protein-DNA complex

The NTD2 domains of the shTnsB1-2 subunits are connected to the DDE domains by a long loop. The intertwined conformation of these protomers with the DNA allows shTnsB1, which recognizes the DNA in LTR(8), to catalyse the strand-transfer reaction on the LTR(6) end and vice versa for shTnsB2 (Figs. 1c, 2b). This crossed positioning of the DDE catalytic domains is favoured by the conformation of the shTnsB3-4 molecules, which stabilize the assembly by intercalating a long bipartite helix that assembles the NTD2 and the DDE domains of different subunits. The adjacent MD accommodates the 5' of the NTS (Fig. 1d, Fig. 3a–c). The C-terminal helix of the OD is snugly fitted between the DDE domain of shTnsB2 and the NTD2 of shTnsB1 by a network of contacts combining polar and non-polar interactions (Fig. 3b), being the former at the end (E360-K520), middle (R367-D512-Q508) and initial (H272-S505) regions of the helix. Another strong polar interaction is observed between the side chains D514 in the OD and R137 in the NTD2. A short loop joins this section with the OD N-terminal helix. The interaction of the second

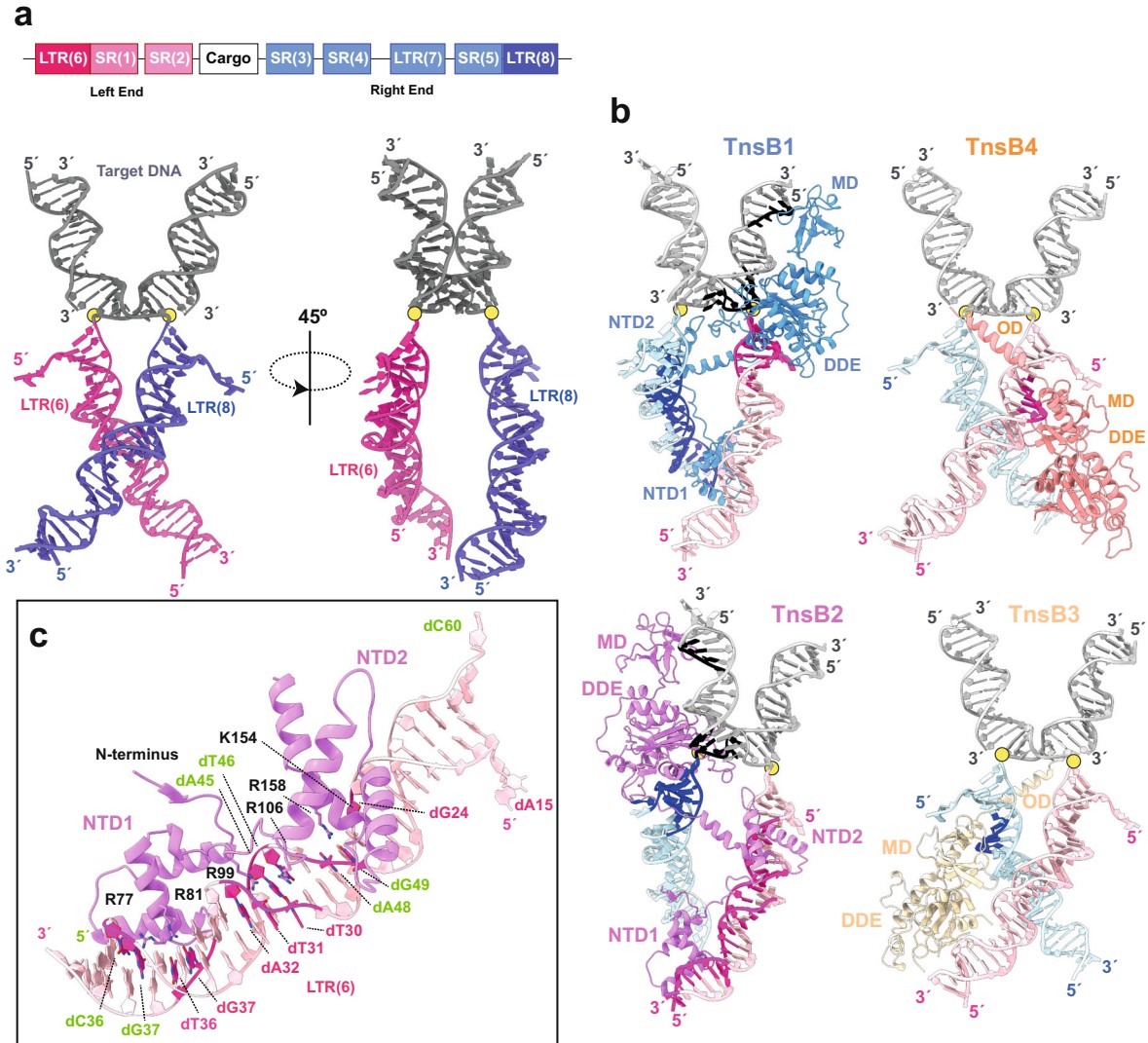

**Fig. 2 | Architecture of the shTnsB-STC complex. a** The upper panel shows a scheme of the *S. hofmannii* transposable element with the left and right ends (LE and RE) at each side of the DNA cargo. The strong colour tone shows the region of each element present in the structure. The lower panel omit the protein moieties in the structure and shows the STC DNA. The DNA is coloured according to the upper scheme for the LE and RE, while the region of the target DNA is shown in grey. The yellow dots indicate the site where the direct attack of the 3′-OH on-target DNA has occurred to promote the strand-transfer reaction. **b** Detailed view of the shTnsB protomer association with the STC DNA. Two different associations are observed for the shTnsB molecules. The different sections of the DNA interacting with the proteins are shown in the corresponding strong colour tone. **c** Detailed view of the helical NTD1 and 2 domains showing the specific interactions with the DNA bases in the LTR(6) region. The labelling of the nucleotides follows the colour code in Fig. 1d, e (see also Fig. 1d and Supplementary Fig. 8).

helix of the OD with the DDE domain is stabilized by non-polar interactions, while a group of residues in the MD of shTnsB3 (R416, Q425, N428) together with R174 and R179 in shTnsB2, form an electropositive charged region stabilizing the bases of the 5′-overhang (Fig. 3c). Finally, the MD and DDE domains of the shTnsB3 protomer associate with the NTD1 of shTnsB2, mainly by a group of non-polar interactions. Interestingly, the N-terminal β-strand of shTnsB2 is inserted as an additional strand in the antiparallel β-sheet of the MD (Fig. 3d). Additional side chain polar interactions (E422-K169, N462-K102 and E54-H283) complete this association. Overall, the associations described here are very similar in the case of shTnsB4 with shTnsB1 and 2. We note that no interaction is observed between shTnsB3 and 4 in the shTnsB-STC complex.

Collectively this network of interactions along the X-shaped complex suggests that the different conformation of the shTnsB1-2 and shTnsB3-4 protomers favours an in trans architecture, most likely as a strategy to link the protein-DNA complex assembly with the DDE catalysis.

## The DDE catalytic domain

Transposases from several superfamilies possess a catalytic domain containing an acidic amino acid triad (DDE or DDD)[25,26]. This protein module catalyses the transposition reaction, whereby the element is excised from the donor site and inserted in the genome or in a mobile genetic element. Mutations in these catalytic residues have shown their critical role in transposition in the DDE domains of Tn5 and Tn10[27,28] transposases. Integration of the elements into a new genomic location usually generates a short target-site duplication from host sequences (2–10 bp). In shTnsB the DDE motif consists of two aspartic acid residues (D205, D287) and a glutamic acid (E321) residue, located in a conserved core that forms a characteristic RNase H-like fold combining α-helices and β-strands (Fig. 4a, Supplementary Fig. 8e). The 3D structure of the DDE triad forms a catalytic pocket which associates with divalent metal ions that assist in the various nucleophilic reactions during DNA cleavage. The DDE domains of the four shTnsB molecules in the assembly superimpose very well (rmsd 0.59 Å rmsd over 160 Cα). However, the DDE pocket displays two different

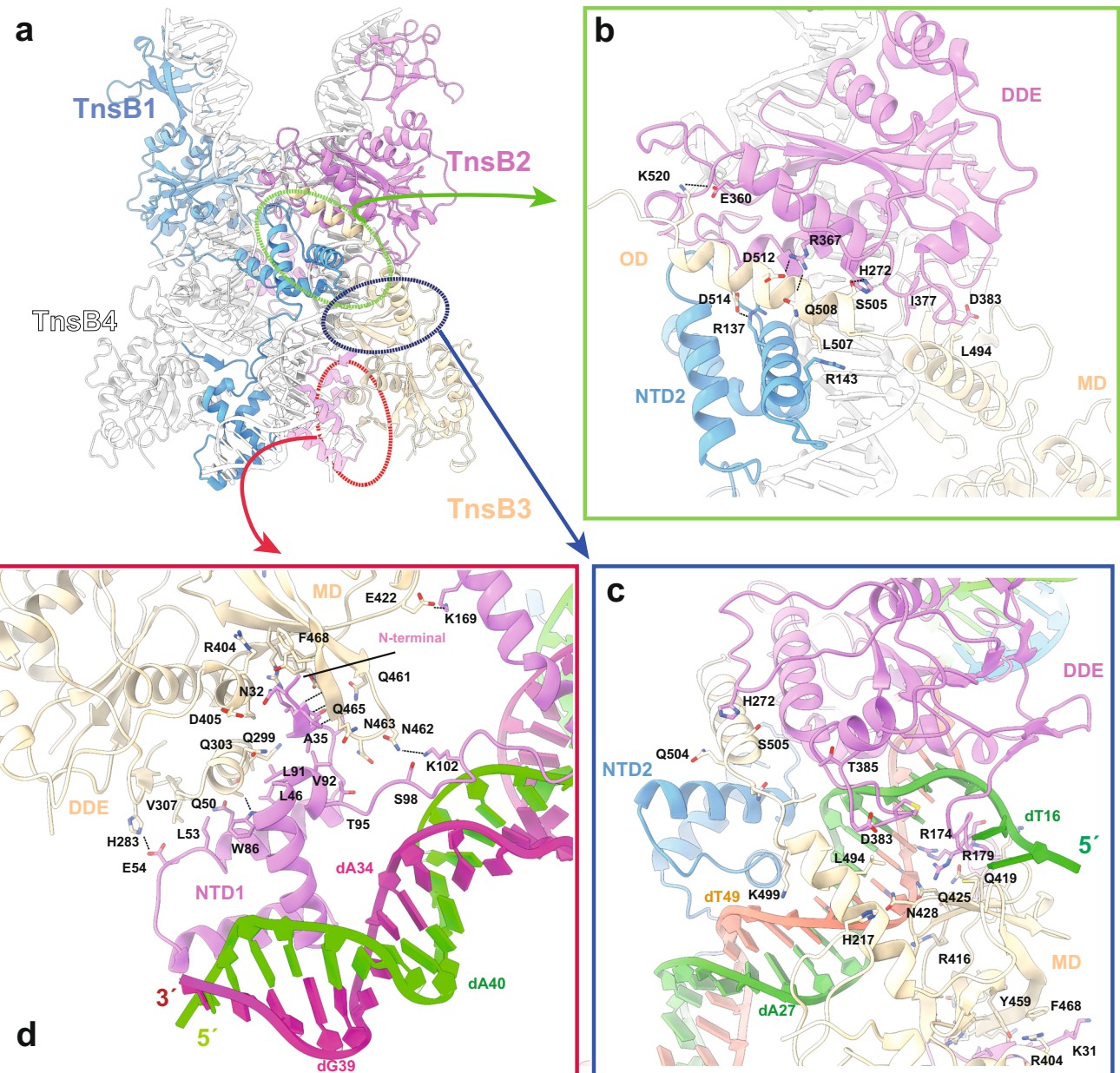

**Fig. 3 | Assembly of the shTnsB protomers. a** Overview of the architecture of the shTnsB protomers and the interactions between the two types of conformations observed for the protein on one of the branches of the STC DNA (transparent). The coloured ovals indicate the zoomed regions in the other panels. **b**, **c** Detailed view of the interactions of the OD of shTnsB3 with the NTD2 and DDE domains of shTnsB1 and 2, respectively. **d** Zoom of the LTR(6) region depicting the association of the N-terminal β-sheet of NTD1 in the MD of shTnsB3.

arrangements depending on the conformation of the shTnsB protomer in the STC complex (Fig. 1). The DDE pockets of shTnsB3 and 4 do not contact the DNA and the catalytic residues of these subunits are not properly positioned for the cleavage reaction (Fig. 4b). In the case of shTnsB1 and 2, they display a DDE catalytic pocket after the strand-transfer reaction has been accomplished. An extra density in the pocket can be visualized (Fig. 4a), which could be assigned to $H_2O$ or $Mg^{2+}$. E321 is 6 Å away, as the structure is captured in the post-catalytic state. The distances of the extra density to D205, D287 and the backbone phosphates suggest that the density could be assigned to the $H_2O$ molecule generated after the 3′-OH attack of one of the strands of the transposable element to the target DNA (Figs. 1a, 4a, Supplementary Fig. 8).

Despite sequence differences, the recent structure of the Tn7 transposon ecTnsB transposase[18] end recognition complex provides a functional homologue for comparisons. The DDE domain is the best-

conserved module of shTnsB, and a search for shTnsB DDE homologues using DALI[29] revealed the domains of ecTnsB and the phage Mu transposase as the closest structural homologues, together with different viral integrases (SRV, HIV-1, HTLV-1). The comparison of the two shTnsB conformations with these proteins showed that the core of the domain superimposed very well with other integrases (Fig. 4c), but only the catalytic pocket of shTnsB1-2 showed the acidic amino acid triad at the appropriate distances to catalyse the strand-transfer reaction (Fig. 4d). In the rest of the cases, the catalytic residues were not properly positioned. Noteworthy, the ecTnsB protomers in the end recognition complex displayed a conformation similar to shTnsB3 and 4 protomers in the STC. However, in the latter, the DNA-binding domains of these two subunits were not visualized (Figs. 1–2), suggesting that the high flexibility observed in the distal sections of the STC DNA (Supplementary Movie 1) disturb their binding to the SR(1) and SR(5) regions of the LE and RE.

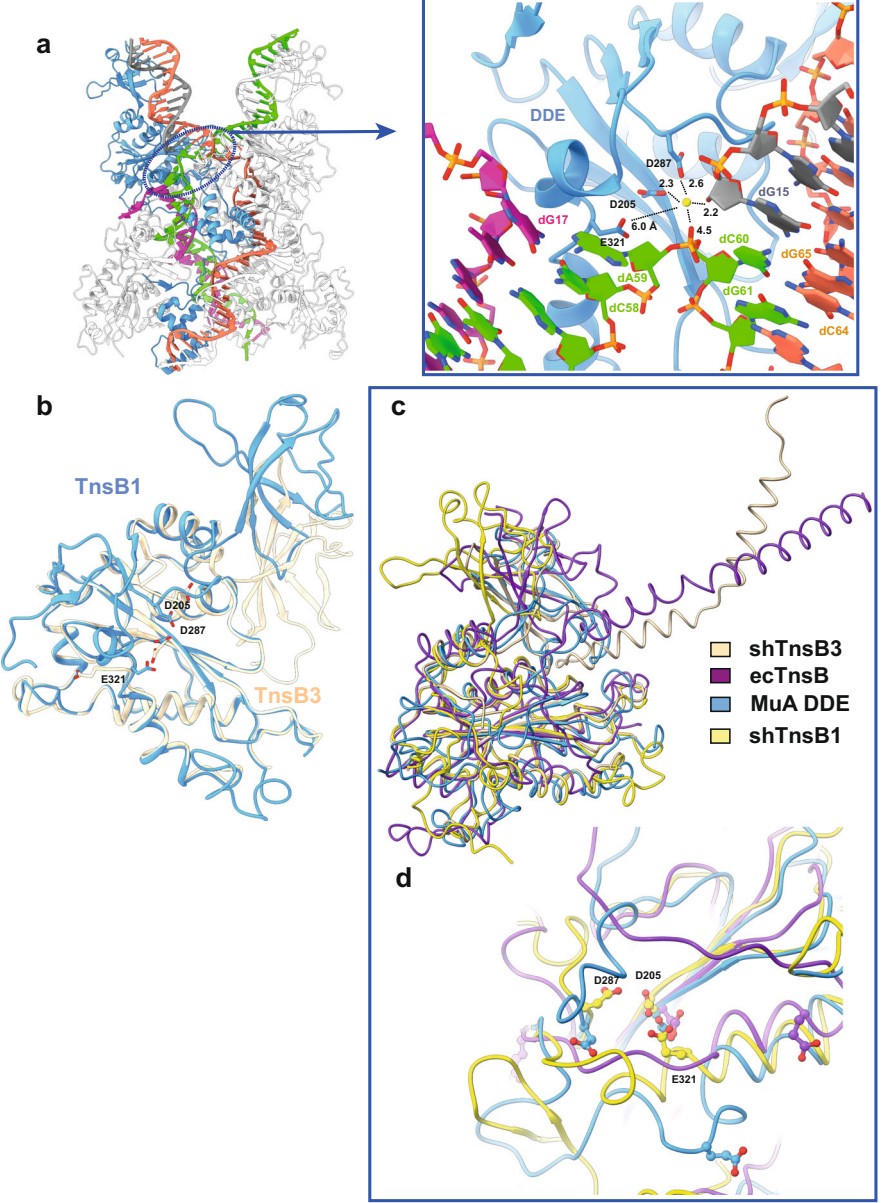

**Fig. 4 | The DDE catalytic site of shTnsB. a** The encircled region locates one of the catalytic centres in the shTnsB-STC complex. The zoom of the region shows the conformation of the catalytic site after catalysis. The putative water molecule and the distances to the neighbouring catalytic residues and the DNA are shown in Å. **b** Superposition of the DDE domains of the two conformations found in the protein-DNA complex for shTnsB. **c** Superposition of the DDE domains of shTnsB2 and 3 with ecTnsB (PDB: 7PIK) and the MuA core domain (PDB: 1BCO). The RMSDs for the superposition are in the range between 0.7 and 1.2 Å for 160 and 250 Cα. **d** The lower panel shows that despite the high structural similarity of the DDE domains, the only subunit in all these structures that displays a competent catalytic pocket is shTnsB2, which is the only one associated with DNA.

Collectively, the structural comparison suggests that the DDE domain maintains a very well-conserved 3D structure; however, the comparison of different structural homologues suggests that the architecture of the catalytic pocket of the transposase is properly arranged once it associates with the target DNA.

## DNA binding by shTnsB, ecTnsB and MuA transposases
The shTnsB protein display homology to ecTnsB and MuA transposases. However, besides the DDE domain, few residues are conserved between them (Supplementary Fig. 1). These differences are especially evident in the DNA-binding domains where multiple sequence alignments from several canonical Tn7 TnsB proteins and diverse classes of CAST elements have shown substantial divergence[18]. The NTD1 of shTnsB shows significant 3D similarities with the HTH domain of the DBD1 in ecTnsB (RMSD 2.4 Å for 66 Cα) and the Iβ domain of MuA

(RMSD 2.8 Å for 62 Cα). Nonetheless, the presence of the N-terminal β-strand (N32-T36) is exclusive of shTnsB, and this region plays an important role in building the assembly of the STC complex (Fig. 3d). This structural feature is not observed in the MuA-STC complex[22], the only structure of a prokaryotic transposase STC complex available so far (Fig. 5).

Intriguingly, the NTD2 domain of shTnsB does not show 3D similarity to its counterpart domains in ecTnsB and MuA transposases (Fig. 5). Instead, NTD2 superimposes well with Pax6, a transcription factor containing a bipartite paired DNA-binding domain (RMSD 2.8 Å for 60 Cα), which has critical roles in the development of the eye, nose, pancreas and central nervous system[30]. The extended linker joining NTD1 and 2 is partially visualized in ecTnsB, and it is not observed in the MuA DNA-binding domains (Fig. 5). This linker makes minor groove contacts, and the carboxy-terminal helix-turn-helix unit makes

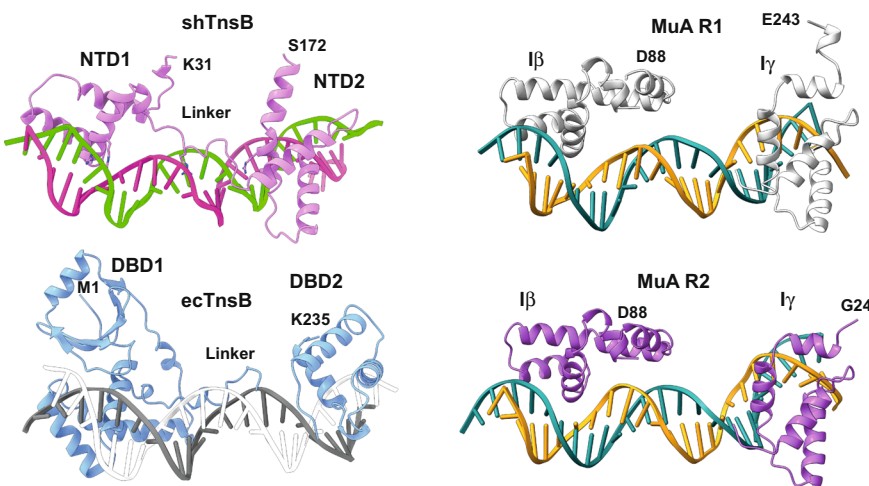

**Fig. 5 | Comparison of the DNA-binding domains of shTnsB, ecTnsB and MuA.** The DNA-binding domains of the three different transposases are shown aligned on the DNA (PDB: 8AA5, 7PIK and 4FCY, respectively). In the case of MuA the same domains bound to the different repeats are shown. For shTnsB only the domains of the protomers 1 and 2 were built. In the case of ecTnsB the DNA-binding domain of one of the subunits in the transposon end complex is depicted.

base contacts in the major groove, as observed in the shTnsB-STC complex (Fig. 5).

Together, the comparison suggests that the TnsB proteins share a common overall DNA-binding mode, including two domains that recognize the different SRs and LTRs in the assembly. However, the domains involved in binding undergo different structural variations to accomplish the specific protein-DNA interactions for each transposase. This diversity could be required to accommodate the sequences of a large number of Tn7-like elements, preventing transposases from recognizing DNA sequences belonging to other transposons[31].

## The STC complexes

The ecTnsB transposon end complex structure has provided information on the recognition of the transposon ends[18], but so far, there is no structural information on their STC complex. The conformation of the ecTnsB subunits in the transposon end complex resembles the arrangement adopted by the DDE domain of the shTnsB3 and 4 protomers in the shTnsB-STC complex. However, the MD and OD domains positions are oriented differently (Fig. 4c). Nonetheless, this conformation is rather different from the architecture of the protomers 1 and 2 of shTnsB, which are involved in the catalysis of the strand-transfer reaction (Fig. 2b). The conformation of these subunits is more similar to its counterparts in the MuA-STC complex (Fig. 6a).

Although the sequence of shTnsB is distantly related to the Mu phage transposase MuA (Supplementary Fig. 1), the general association on the MuA-STC complex shares similitudes with the CAST protein (Fig. 6a). However, the assembly also presents important differences. One of the main differences is that in the MuA-STC complex, the DNA-binding domains of the subunits involved in the assembly (MuA 3 and 4) are bound to the corresponding repeats in the LE and RE (Fig. 6a), while in the shTnsB-STC they could not be modelled, suggesting that they are highly flexible. This notion is supported by the high variability observed in the external part of the X-shaped DNA structure (Supplementary Movie 1). The fact that those regions of the MuA-STC structure[22] are involved in crystal contacts could reduce the flexibility, thus facilitating the binding of the Iβ and Iγ domains to the repeats in the LE and RE ends.

Curiously, the association of the unique NTD1 N-terminal β-strand in shTnsB is not observed in MuA (Fig. 6a, b upper panels), as this secondary structure element is not present neither in the Iβ domain of MuA nor in the DBD1 of ecTnsB. This observation suggests that in the ecTnsB-STC complex, this association cannot occur. Another important difference arises from the assembly of the complex where the OD domain in shTnsB intercalates, with multiple interactions, between the NTD2 and the DDE catalytic domain, building the association between shTnsB proteins. In the case of MuA the association between the four protomers in that region of the STC complex is different, displaying considerably fewer interactions between the different domains (Fig. 6b). Furthermore, the OD domains are not observed in the protomers 1 and 2 of the shTnsB-STC complex, while in the case of MuA the helices in the IIIα domain are crossed in the interior of the V-shaped target DNA region (Fig. 6a). The possibility that the interaction between these helices in the MuA structure is due to crystal contacts could not be disregarded.

## The effect of DNA binding in transposition

To validate the shTnsB:DNA interactions in the STC, we tested the transposition activity of a series of substitution mutants where conserved amino acids displaying polar interactions with the DNA were replaced by alanine (Fig. 7, Supplementary Fig. 1). We mutated R77, R81 (both present in NTD1), R99 (present in the linker between NTD1 and NTD2), R158 (present in NTD2), R188, R223, K290 and R380 (present in the DDE domain) (Fig. 7b). The R77A, R81A and R158A mutants showed a dramatic effect on on-target transposition activity, suggesting that shTnsB can no longer recognize the repeats in RE and LE (Fig. 7b). Noteworthy, the nucleotides involved in this interaction (Supplementary Fig. 7) are conserved between the different repeats further supporting the importance of this interaction in donor DNA recognition[21]. As for R99, even though it is present in a disordered region of shTnsB, it has an intimate interaction with the phosphate backbone and the bases of the LTRs. These nucleotides are also conserved within the RE/LE repeat set[21]. However, the effect of its substitution was variable, with replicates displaying either higher or lower activity than the wild-type. In the case of R223A and R380A, the activity was also affected. These residues seem to be involved in stabilizing the post-transposition state. R223 interacts with the central nucleotides of the 5 bp sequence between the nicks generated by transposition, while R380 interacts with the non-cleaved transposon ssDNA positioned out of the complex. Finally, despite its conservation and proximity to the attachment site, mutants R188A and K290A did not abolish on-target transposition. Our results confirm the functional importance of these residues involved in LTR recognition and STC stabilization and reveal possible sites for improvement of the transposase properties for genome engineering purposes.

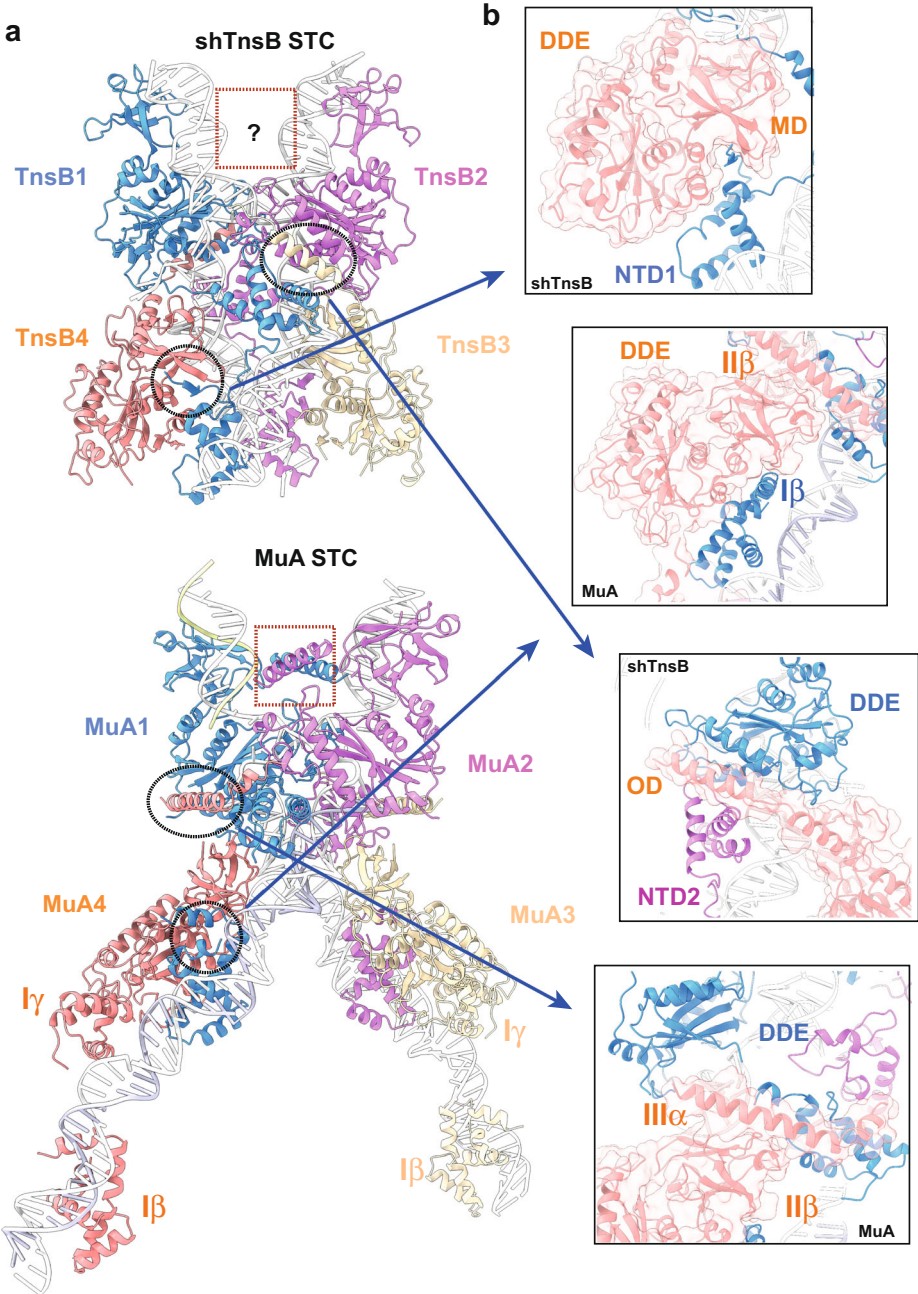

**Fig. 6 | Analysis of the strand-transfer complexes of shTnsB and MuA. a** The shTnsB-STC and MuA-STC complexes are shown following the same colour scheme as in Fig. 1, except for the DNA which is coloured in white. **b** The encircled regions on both complexes are compared to show key differences in the assembly of the STC complexes.

## Discussion

Last year witnessed a large increase in structural knowledge of CAST systems. Structural information of TnsC, TniQ and Cas12k of type V-K[32–34], as well as for type I-F Cascade complex bound to TniQ[35,36] is available. However, despite these advances, we need precise insight into the sequence of interactions and intermediates leading to precise DNA cargo insertion to address possible engineering of the CAST systems. This lack of thorough understanding of how transposition is accomplished by the different CASTs, precludes their repurposing for precise insertion of DNAs into gene-editing tools. Type V-K could represent a promising system for the development of these tools due to the reduced number and size of its components. However, the generation of co-integrates must be addressed for this purpose (Fig. 1a).

The low-resolution shTnsB transposon end complex (Supplementary Fig. 3) suggests that the transposon end recognition is similar to the ecTnsB[18]; however, the high heterogeneity and flexibility of the assembly did not allow a detailed molecular characterization of the DNA recognition. This agrees with experiments in the ecTn7 system, which have shown that the pre-transposition complex is less stable than the post-transposition complex and that transposon protection changes between the pre- and post-catalytic stages[17,37]. In addition, EMSA experiments suggest that shTnsB can interact with the RE and LE in the pre-catalytic state in the absence of the target DNA (Supplementary Fig. 2). Thus, shTnsB interaction with the transposon ends seems to be independent of the presence of other components from the type V-K CAST system. The higher stability of the post-catalytic complexes led us to assemble the shTnsB-STC, which we were able to

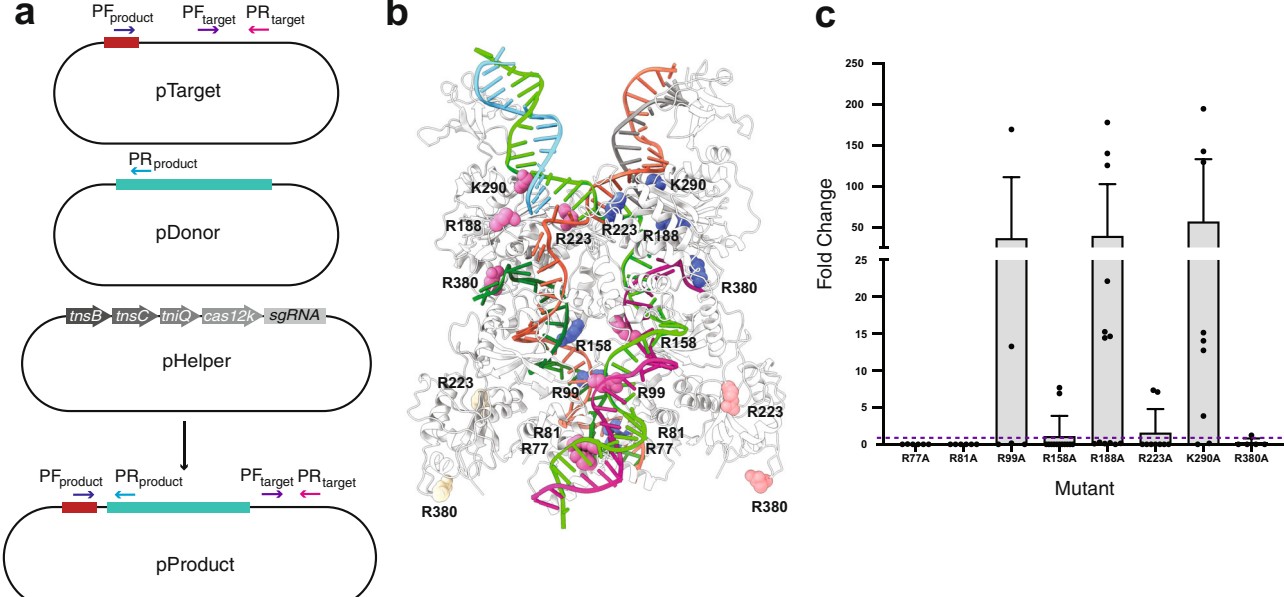

**Fig. 7 | In vivo transposition assay and qPCR quantification. a** Transposition was carried out by transforming bacteria with a plasmid containing a target site, shown in green, recognized by Cas12k:sgRNA (pTarget), a plasmid containing a donor DNA with the CAST RE and LE sequences shown in red (pDonor) and a plasmid comprising the sgRNA, Cas12k, TnsC, TniQ and TnsB coding sequences (pHelper). Integration of the donor sequence into pTarget was quantified by qPCR using a set of primers amplifying a region of pTarget where no transposition occurs ($PF_{target}$ and $PR_{target}$) and a set of primers that amplify a region between the target and the donor LE sequences ($PF_{product}$ and $PR_{product}$). **b** Position of selected amino acids within the STC to be mutated and tested for transposition activity. **c** Activity fold-change measured by qPCR for each TnsB mutant. Two or three biological replicates were performed for each mutant. Individual data points representing all technical replicates are depicted as dots. Bars represent the activity fold-change mean, and whiskers centred around it represent the standard deviation. The wild-type activity is represented by the dashed line. Data are provided as a Source Data file.

determine at high resolution. This complex represents the stage after completion of the transesterification reaction with the target DNA, revealing its 3D assembly. The overall structure resembles the architecture of the phage MuA transpososome[22]. Yet, key differences between the assemblies can be observed, especially in the NTD2 domain (Fig. 6). One similitude between the shTnsB and MuA, which differentiates them from the Tn7 system, is that the Type V-K and Mu use replicative transposition, which requires the cleavage of only one DNA strand, while Tn7 performs a cut-and-paste reaction, involving the cleavage of the non-transferred strand by TnsA. Therefore, the structural similarities of the overall STC complex between shTnsB and MuA would be supported by these mechanistic similitudes. Yet, it seems difficult to establish analogies of the possible STC complex assembly of ecTnsB[18] using the shTnsB-STC structure, as we will expect that ecTnsB-STC should also accommodate ecTnsA[10]. The same could be extrapolated for type I-F and I-B CAST, which include TnsA in their composition.

The post-catalytic shTnsB-STC structure revealed very few base-specific protein-DNA contacts (Supplementary Fig. 7), as it has been also observed in the ecTnsB pre-catalytic complex[18]. The NTD1 domain of shTnsB partially shares its architecture with its counterparts in ecTnsB and MuA, while NTD2 is unique in the sense that it shares structural homology with the Pax6 transcription factor (Fig. 5). The N-terminal β-strand linking the recognition in one of the transposon arms with the catalytic centre on the opposite side and vice versa is a singular feature of shTnsB, whether this feature exerts some allosteric regulation in the DDE catalytic site remains to be determined. The two subunits of shTnsB engaged in the reaction display the acidic triad in a post-catalytic state. The attack of the 3′-OH of the transposon strands on the 5′ of the target DNA leaves uncoupled bases (dT62 and dA63) in each strand and a water molecule bound to the aspartic residues and the DNA backbone (Supplementary Fig. 7). Based on the structures of the shTnsB-STC and the ecTnsB transposon end complex structures,

we can conclude that the active pocket of the TnsB proteins is not arranged for catalysis before the target DNA is bound (Fig. 4d). Most likely this mechanism is combined with the trans assembly of the STC to preclude non-specific cleavage.

ShTnsB inserts the transposon with a directionality bias[16]. Although the LTR8-SR5 and LTR6-SR1 ends are almost symmetrical, the length and the sequences from the right and left ends are different (Supplementary Fig. 2f), suggesting that these differences, combined with the need for a PAM to determine the insertion site, could influence the directionality of the cargo DNA. This notion is supported by the observation that the distance between the PAM and the transposon attachment site shows a very narrow distribution[16]. However, the mechanism governing DNA cargo integration is not fully understood. Two alternative models have been proposed[32,34]. In both cases, the assembly of the shTnsC filament would work as a molecular ruler to direct shTnsB to the attachment site. Indeed, the C-terminal region of shTnsB interacts with shTnsC[32,34], and the shTnsB-shTnsC interaction results in filament disassembly. However, one model suggests that the shTnsC filament grows from shCas12k, and the association of the head of the filament with shTniQ would indicate the insertion site for shTnsB and the DNA cargo[32]. The second model proposes that the shTnsC filament grows in the opposite direction, and the shCas12k-shTniQ complex will stop the growth recruiting shTnsB with the DNA cargo, thus proceeding with the integration[34]. Both models pose additional questions regarding the shTnsB-shTnsC interaction. As shown in the shTnsB-STC structure, not all protomers display the same conformation, and it is unclear whether all shTnsB protomers, the catalytically active or the inactive protomers interact with shTnsC (Fig. 1d, e), as in both cases the C-terminus is not visualized in the structure. However, it seems to be topologically available for additional interactions. In the light of the current biochemical and structural data[32,34], we would like to propose an alternative hypothetical working model (Fig. 8), which considers that shTnsC and shTnsB would be associated before shTnsC

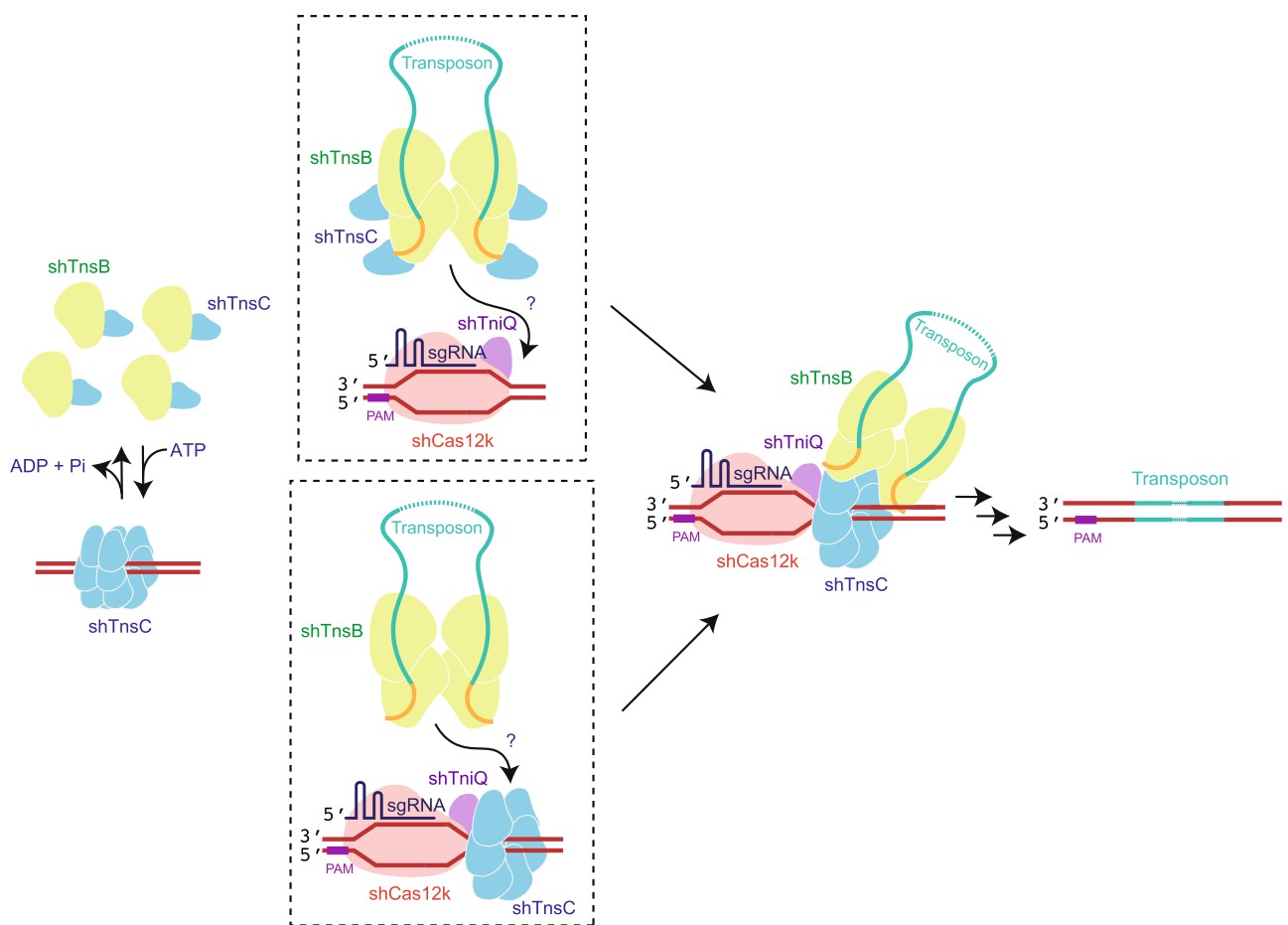

**Fig. 8 | Alternative mechanism of RNA-guided integration in type V-K CAST system.** ShTnsC binds DNA forming filaments that are depolymerized by shTnsB (left). Given shTnsC interactions with shTnsB and the shCas12k-shTniQ, two possibilities arise (dashed squares). First, shTnsC bound to shTnsB could be recruited by the targeting complex comprising shCas12k and shTniQ (top dashed square). Second, shTnsC could bind the targeting complex first and recruit shTnsB afterwards (bottom dashed square). Both possibilities would produce a pre-transposition complex activating shTnsB integrase activity and resulting in transposon insertion.

interacts with DNA, to avoid unspecific interactions with the genome, which could be toxic for the cell. Thus, the formation of the shTnsC filaments would be dynamic, as the presence of shTnsB would induce the dissociation of the assembly in the absence of shCas12k and shTniQ. In this scenario, the association of shCas12k with the target DNA and shTniQ will stabilize the assembly of the shTnsC-shTsnB complexes in the insertion site, then triggering the integration of the DNA cargo. Further analysis of the shTnsB-shTnsC association will be needed to fully understand the DNA cargo integration in type V-K CAST and validate any of these working models.

The absence of TnsA in type V-K leads to further questions regarding integration intermediate resolution since non-cleaved transposon 5' ends result in cointegrate transposition products[15,16]. Transposons relying on DDE transposases have different mechanisms to solve this problem[26,38,39]. For instance, Tn5053 contains resolvase TniR, which uses *res* sites, also present in the transposon, to resolve the cointegrate[40,41] while other transposases generate hairpins to excise the remaining DNA[42]. However, the lack of the insertion domain that stabilizes the hairpin in the catalytic DDE domain of shTnsB does not support this possibility[16] to prevent co-integrates. Additionally, DNA replication and homologous recombination can solve the cointegrate in the case of replicative transposition[26].

On the other hand, the presence of a host-specific resolvase is supported by the fact that when transposition is carried out in *E. coli*, 20% of the insertion products are co-integrates[16]. Indeed, the V-K system geneset (TnsB, TnsC and TniQ) is similar to that of Tn5053's, which contains TniA, TniB (TnsB and TnsC homologs) and TniQ[16], and it could be hypothesized that a resolvase not encoded in the V-K transposon locus but still present in the natural host could have the same activity as resolvase TniR. Further studies would need to be performed in *S. hofmannii*, or other related cyanobacteria, before we can fully understand RNA-guided transposition in Type V-K. Solving these questions is key to successfully generating new engineered CAST complexes able to integrate large DNA cargoes in eukaryotic genomes.

Our structure provides a mechanistic understanding of how CAST shTnsB recognizes the DNA sequences of the transposon RE and LE in the STC. The structure also shows the conformational gymnastics of the shTnsB protein to assemble with the DNA and provides a detailed view of the active site after catalysis. Full elucidation of the assembly mechanisms of shTnsB with the rest of the CAST components will be required to harness CAST systems for RNA-guided genome engineering.

## Methods

### Protein production and purification

The *Scytonema hofmannii* (UTEX 2349) *tnsB* coding sequence (IDT) was cloned with a *C*-terminal hexahistidine (His)-tag into pET-21 vector (Supplementary Table). Proteins were expressed in *E. coli* BL21 pRARE cells. *E. coli* cultures were grown at 37 °C in Terrific Broth (TB) medium with 100 mg/L ampicillin to an $OD_{600}$ of 0.6. Overexpression of proteins was induced with 250 mM of IPTG for 16–18 h at 18 °C. Cells were harvested by centrifugation, flash-frozen in liquid Nitrogen and stored

at −80 °C. Pellets were subsequently resuspended in lysis buffer (50 mM HEPES pH 7.5, 500 mM NaCl, 5 mM MgCl$_2$, 1 mM TCEP, 1 tablet of Complete Inhibitor cocktail EDTA Free (Roche) per 50 mL, 50 U/mL Benzonase, 2 mg per 50 mL lysozyme). The resuspended pellets were stirred for 1 h at 4 °C. Lysis was completed by sonication, and lysate was centrifuged at 10,000 × $g$ for 45 min to separate the soluble and insoluble fractions. The soluble fraction was loaded into a 5 mL HisTrap FF Crude column (Cytiva) equilibrated in buffer IMAC-A (50 mM HEPES pH 7.5, 500 mM NaCl, 1 mM TCEP, 10 mM imidazole, 1 tablet of Complete Inhibitor cocktail EDTA Free (Roche) per 500 mL), and bound proteins were eluted by the stepwise increase of the imidazole concentration with buffer IMAC-B (50 mM HEPES pH 7.5, 500 mM NaCl, 1 mM TCEP, 500 mM imidazole, 1 tablet of Complete Inhibitor cocktail EDTA Free (Roche) per 500 mL). Enriched protein fractions were pooled together, and NaCl concentration was reduced to 200 mM using dilution buffer (50 mM HEPES pH 7.5, 50 mM NaCl, 1 mM TCEP). The sample was applied onto a HiTrap Heparin HP column (GE Healthcare) equilibrated with buffer Heparin-A (50 mM HEPES pH 7.5, 200 mM NaCl, 1 mM TCEP). The protein was eluted with a linear gradient of 0–100% buffer Heparin-B (50 mM HEPES pH 7.5, 1 M NaCl, 1 mM TCEP). Fractions containing the protein were pooled and mixed with TEV-protease to cleave the his-tag. Then, the protein was concentrated and further purified by size exclusion chromatography (SEC) using a HiLoad 16/600 Superdex 200 column (Cytiva) equilibrated in SEC buffer (50 mM HEPES pH 7.5, 500 mM NaCl, 1 mM TCEP). Fractions containing pure protein were pooled, concentrated to 9 mg/mL, flash-frozen in liquid nitrogen and stored at −80 °C.

### DNA production
DNA sequences corresponding to individual repeats, double repeats and STC ssDNA were purchased from IDT. DNA molecules containing three or more repeats were obtained by PCR with a donor DNA cloned into pUC57 (Genewiz) as the template (Supplementary Table 1). The DNA sequences were selected from the natural transposon for Type V-K CAST in *S. hofmannii* described in[4,7,21]. Complete donor DNA was amplified using primers 5′-tgctacgtctctacgtgtacagtgactaat-3′ and 5′-ccacaaaaggcgtagtgtacagtgacaaat-3′, RE was amplified using primers 5′-tgctacgtctctacgtgtacagtgactaat-3′ and 5′-atatacaaagcgacagtcaatttgtcattatgaaaat-3′ and LE was amplified using primers 5′-attatattgatgacatttaatttgtcatcaattaattaag-3′ and 5′-ccacaaaaggcgtagtgtacagtgacaaat-3′. The amplified sequences were purified using QIAquick PCR Purification kit (Qiagen) and eluted in nuclease-free water.

### Electromobility shift assay (EMSA)
DNA (donor, RE, LE or LTR(6)-SR(1)) was mixed with shTnsB in buffer containing 50 mM HEPES pH 7.5, 300 mM NaCl, 5 mM MgCl$_2$ and 1 mM TCEP. DNA and shTnsB concentrations are indicated in Supplementary Fig. 3c–e. Samples were incubated at 37 °C for 30 min and loaded onto 6% DNA retardation gels (Thermo Fisher Scientific). DNA was detected by staining the gel with GelRed (VWR). The same gel was afterwards stained with InstantBlue (Life Technologies) to detect the presence of shTnsB and compare band shifts.

### Cryo-EM sample preparation
The pre-catalytic complex was prepared using 1 μM of RE, 1 μM of LE and 8 μM of shTnsB in 300 μL of buffer containing 50 mM HEPES pH 7.5, 300 mM NaCl, 5 mM MgCl$_2$ and 1 mM TCEP. The sample was incubated for 30 min at 37 °C and concentrated to 50 μL in 10 kDa filters. The STC reconstitution is depicted in Supplementary Fig. 4 and was performed as follows. The transferred and non-transferred strands (TS and NTS) from RE (RE_Target and RE_polyA oligonucleotides in Supplementary Table 1) and LE (LE_Target and LE_polyA oligonucleotides in Supplementary Table 1) were mixed separately, incubated at 95 °C for 5 min and left on ice for 10 min. Subsequently, the complementary target ssDNA was added to the TS-NTS dsDNA from RE and

LE (Target_1 and Target_2 oligonucleotides in Supplementary Table 1 correspondingly). The TS-NTS dsDNAs from RE and LE were incubated separately at 70 °C for 5 min and left on ice for 10 min. Then, they were mixed and incubated at room temperature for 10 min. Finally, 50 mM HEPES pH 7.5, 300 mM NaCl, 5 mM MgCl$_2$ and 1 mM TCEP were added to 10 μM DNA STC DNA before adding 40 μM shTnsB. The sample was incubated for 30 min at 37 °C.

Grids for both samples (pre-catalytic complex and STC) were prepared by applying 3 μL of freshly prepared complex (1:2 dilution) to UltrAuFoil 300 mesh R1.2/1.3 holey grids (Quantifoil), glow-discharged for 60 s at 10 mA (Leica EM ACE200) and plunge-frozen in liquid ethane (pre-cooled with liquid nitrogen) using a Vitrobot Mark IV (FEI, Thermo Fisher Scientific−blotting time of 3 s, 100% humidity, 4 °C).

### Cryo-EM data processing
Data collection was performed using EPU 2.12.1 (ThermoFisher). Data processing was carried out using cryoSPARC version 3.3.2[23]. Patch motion correction and patch CTF estimation was performed. 9646 K particles were initially picked from 4574 micrographs using the blob picker with min and max diameters of 50 and 200 Å, respectively, and subsequently extracted and sorted by two rounds of 2D classification. The selected 2D classes contained 415 K particles, extracted using a 416 × 416 pixel box. An ab initio 3D reconstruction was used as a starting volume for non-uniform refinement, and the resulting particles were analysed by 3D variability analysis[43] (Supplementary Fig. 5), which revealed a continuous conformational variability in the distal ends of the DNA and nearby shTnsB domains (N-terminal part and CTD of chains A and B). Two of the most spatially different volumes produced were used as references for heterogeneous refinement and were further refined by non-uniform (NU) refinement to a global resolution of 2.47 Å (269 K particles) and 2.81 Å (155 K particles), based on the gold-standard Fourier shell correlation (GSFSC) 0.143 cutoff criterion (Supplementary Fig. 6). Per-particle defocus, and per-group CTF parameters were refined during the NU refinement. On visual inspection, the density of the 2.47 Å map appeared considerably more continuous. This map was further improved in the distal ends by another iteration of 3D variability analysis of the particles[43] and using the most spatially different volumes as references for another round of heterogeneous refinement. Most particles ended up in one class (208 K) which were finally refined by NU refinement to a 2.46 Å map. 3DFSC analysis revealed sphericity of 0.928 and an angular resolution range of 2.4–3.1 Å (Supplementary Fig. 6). Local resolution analysis was performed with MonoRes[44] using the cryoSPARC wrapper, which revealed a range of ~2–10 Å with the majority being around 2.5 Å, and distal part being 4-8 Å. Model building was performed in maps that had been post-processed using DeepEMhancer 0.13[45]. Half-maps were used as input, while normalization and masking/denoising of the input volumes was performed automatically in the tightTarget mode, yielding maps of significantly improved interpretability. The final model built in the DeepEMhancer maps was finally cheeked using the experimental maps.

### Atomic model building and refinement
An initial model was generated based on the MuA structure (PDB: 4FCY), using the automated protein structure homology-modelling server, SWISS-MODEL[46]. It was rigid body fitted into the map using ChimeraX[47]. The model fit was inspected and further fitted in COOT[48], and less ordered N- and C-terminal regions were removed. An Alphafold[49] model of the shTnsB monomer was subsequently used to fit some of these regions into the map as well. The model was further rebuilt de novo in COOT, and refined using phenix.real_space_refine[50], while the Namdinator molecular dynamics flexible fitting server was used to speed-up and improve the model to map fit and for model refinement in both early and late stages of refinement[51].

## In vivo transposition assay and mutagenesis

CAST transposition activity was detected using vectors pHelper, pTarget and pDonor obtained from Addgene (Cat# 127922, 127926 and 127924). shTnsB mutants (Fig. 7) were generated in the pHelper vector using the In-Fusion cloning kit from Takara and primers containing the desired point mutations. In vivo assays were performed following Saito et al.[52]. Briefly, 12 ng each of pHelper, pTarget and pDonor were transformed into 30 μL of TransforMax EC100 pir+ Electrocompetent *E. coli* (Lucigen). Cells were recovered for 1 h and plated on LB media containing ampicillin, kanamycin and chloramphenicol. Cells were scraped 24 h after plating, lysed in 15 μL colony lysis buffer (TE with 0.1% Triton X-100), boiled for 5 min, diluted with 30 μL of water and spun at 4000 × *g* for 10 min to pellet debris. The supernatant was used for subsequent analysis.

## Activity assay qPCR

The frequency of insertions relative to the concentration of the target plasmid was determined by qPCR (TaqMan Fast Advanced Master Mix, Applied Biosytems) with a set of primers/probe (5′-6-FAM, Int ZEN, 3′-Iowa Black, IDT) corresponding to an unmodified region of pTarget (primer forward: 5′-cgacagcatcgccagtcactatg-3′, primer reverse: 5′-caagtagcgaagcgagcaggac-3′, probe: 5′-tgcgttgatgcaatttct atgcgcacccgt-3′) and a set of primers/probe (5′-6-FAM, Int ZEN, 3′-Iowa Black, IDT) corresponding to the transposition product (primer forward: 5′-ggttgagaagtcatttaataaggccactgttaaacg-3′, primer reverse: 5′-aacgctgatgggtcacgacg-3′, probe: 5′-ctgtcgtcggtgacagattaatgtc attgtgac-3′) (Fig. 7). Each reaction (11 μL) contained 2 μL of extracted nucleic acids, 900 nM forward primer, 900 nM reverse primer and 250 nM of probe. Fluorescence was measured using the LightCycler 480 Instrument (Roche). Data were analysed by the $2^{-\Delta\Delta Ct}$ method normalizing with pTarget Ct.

## Reporting summary

Further information on research design is available in the Nature Research Reporting Summary linked to this article.

# Data availability

Reagents generated in this study are available upon request with a completed Materials Transfer Agreement. The atomic coordinates and cryo-EM maps included in this study have been deposited in the Protein Data Bank and Electron Microscopy Data Bank under the accession codes: PDB 8AA5, EMD-15294 and EMD-15344. qPCR source data and uncropped gels are provided with this paper. Source data are provided with this paper.

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

## Acknowledgements

We thank the Danish Cryo-EM National Facility in CFIM at the University of Copenhagen and specially Tillmann Pape for support during cryo-EM data collection. We also thank the Protein Expression Unit at CPR for assistance in protein expression and purification. F.T.C. is a member of the Copenhagen Bioscience PhD programme (NNF19SA0035440). G.M. is part of the Novo Nordisk Foundation Center for Protein Research (CPR), which is supported financially by the Novo Nordisk Foundation (grant NNF14CC0001). This work was also supported by grant NNF0024386, grant NNF17SA0030214 and Distinguished Investigator (NNF18OC0055061) grants to G.M., who is a member of the Integrative Structural Biology Cluster (ISBUC) at the University of Copenhagen.

## Author contributions

G.M. conceived the study, F.T.C. and G.M. designed the biochemical experiments. F.T.C. and S.S. set up the expression and purification protocol. F.T.C. and A.F. characterized the protein-DNA binding properties. F.T.C. and L.S. created the mutants and performed transposition experiments. F.T.C. and B.L.M. performed the SEC-MALS analysis, F.T.C. prepared EM grids and collected the cryo-EM images with N.S. F.T.C. and N.S. performed cryo-EM data processing, built the models and G.M. proceeded with cryo-EM map and structure analysis with their help. The global results were discussed and evaluated with all authors. G.M. coordinated and supervised the project and wrote the manuscript with input from all the authors.

## Competing interests

G.M. and S.S. are co-founders and members of Twelve Bio BoD. The remaining authors declare no competing interests.
