## [Peer Review File · Nature Communications]

Structure of the TnsB transposase-DNA complex of type V-K
CRISPR-associated transposonREVIEWER COMMENTS

Reviewer #1 (Remarks to the Author):

Dr. Montoya and co-workers present a cryo-EM structure of the TnsB DNA transposase complex with its strand-transfer DNA product from the V-K CRISPR associated transposon. The result is novel, as this is the first such complex solved. The main finding of the work is that the TnsB strand transfer complex looks very similar to that of the phage MuA DNA transposase strand-transfer complex, suggesting that the molecular architecture of the complexes indeed reflects the mechanistic relationship between these two transposition systems, as both are the replicative DNA transposons. The difference is very sharp when compared to the recently published cryo-EM structure of ecTnsB. This difference might very well reflect the fact that ecTnsB is part of transposon ecTn7 that is a cut-and-paste element. However, one should also be cautious, as the ecTnsB complex was generated using only transposon ends, so we do not really know what an ecTnsB strand-transfer complex looks like.

Unfortunately, the paper as written is somewhat dull, describing a large amount of structural details without putting them in a biological and/or mechanistic context. The authors do not venture to address the most exciting and outstanding question in the field, which is how V-K CAST TnsB integrates the transposon at a specific site at a defined distance away from the sequence recognized by the crRNA. It is well known that this feature is somehow dependent on the dynamic interaction between TnsB and the key ATPase regulator of the system, TnsC. From the structure and previous work, authors should know where TnsB interacts with TnsC to trigger ATP hydrolysis, I'm surprised that this is not pursued in the Discussion, not even to the extent to suggest that perhaps which of the protomers might interact productively with TnsC. This omission seems especially pointed, given the rapidly emerging structural and mechanistic understanding of TnsC, even in the precise context of V-K CAST transposon systems.

Furthermore, the paper uses terminology in numerous places that is likely to be confusing to both the general reader and those familiar with the field.

Specific Comments:

Line 95. "Tn7-like transposons "parasite". I don't think this is really true, as who is parasitizing whom is a complex question and I'm not aware any data deciding the nature of this relationship. Tn7 is certainly not parasitizing anything for "mobilization" as Tn7 is perfectly capable of mobilizing itself. What can be said is that Tn7 has apparently evolved to employ a number of different site-specific targeting strategies and reliance on a crRNA is one of these.

Line 121. "as it is unknown". In fact, the presence of co-integrates suggests that it is known, and very likely the 5' end is not processed by the transposase, and co-integrates are likely resolved by homologous recombination. Given that the authors have an in vivo assay in hand, they could have relatively easily asked the question whether the ratio of co-integrates changes if transposition was carried out in RecA- cells. One must point out that there are a number of replicative DNA transposon (IS5, IS6 families) that rely on resolution by host encoded systems; therefore, I do not see why something mysterious should be invoked unless data suggest that indeed something else beyond the obviously expected is going on.

Line 154. Please rename left and right element as left and right ends, as is common in the literature. "Element" in the field typically refers to the mobile genetic element or MGE itself.

Line 157. "ladder of discrete bands, which increased" I do not know how a ladder increases; please rephrase.

Line 160. "we tested whether shTnsB can bind independently". Please state what was the result.

Line 174. "TnsB and a RE and LE not bound to each other." Please rephrase, as I have no idea what this means.

Line 184. I do not find the description of the STC DNA construction easy to understand in the Methods. If I understand correctly, the STC DNA consists of six DNA strands, including two long transferred strands (ST). In these the transposon part is linked to the target part through a phosphodiester at the scissile phosphate. The non-transferred strands (NTS) are broken at the transposon/target junction with some bps missing to ensure base pairing, giving four other strands. This is the standard way to assemble STC DNA. Yet the terminology used in Methods speaks only about LE and RE, so it is not clear. I do not understand what "LE and RE ssDNA were mixed separately" means (line 611). Also, please use the TS and NTS terminology as this is common in the field. The most straight-forward might be to give the six DNA strands unique names, and describe from there. Furthermore, should change the "LTR" designation, as this is generally understood to mean something else altogether, denoting Long Terminal Repeats in retro-transposition and retrovirology. It is not clear how the target DNA sequence was chosen, so please describe this clearly. In particular, it is not clear what target site was chosen and why, or whether this particular site was in fact chosen with intent. Really informative is Suppl. Fig6, where the STC DNA is finally shown (although - if I understand correctly - only the visible portion?). I think this figure should come much earlier. I would like to know whether TS target DNA base pairing between the two TSs was intentionally designed as such, and if so why. What is the significance of this base pairing, and what is the significance if any of the apparently flipped out dA63 of the right NTS? Does this base-pairing and/or the flipped out base have any biological significance? Are these present at the biological

target sites? If biologically significant, this could be a rather important finding in itself. I note that the otherwise very useful Fig1d ignores this aspect of the STC design and structure, which is baffling.

In general it appears that the authors have managed to generate complexes with one LE+target and one RE+target bound to it. The authors should explain why symmetrical complexes (LE/LE or RE/RE) bound were not observed and how they know for sure that in their observed complexes both LE+target and RE+target are indeed present.

Line 189. Fig 1a. This figure is not correct and misleading as it is drawn, as it implies that 5' cleavage somehow mysteriously occurred and carried out by shTnsB. There are no data for this. In fact, the likely scenario is that a Shapiro intermediate formed, leading to a co-integrate that is somehow resolved. I'm afraid that a more complicated figure needs to be supplied that describes the likely strand cleavage and strand-transfer events.

Line 220. "inferior ends" I do not know what inferior stands for.

Line 222. "opposite branch" Please rephrase so it is compatible with the field's standard nomenclature.

Line 245. "MyB/homeodomain like fold". Please give appropriate reference to this fold with a brief sentence as to what these proteins do.

Line 246. "well-conserved". Please state that among what proteins R77 is conserved.

Line 253. "structure is not related" Please amend the negative statement with the results of the Dali search and state what this domain is actually related to.

Line 270. "5' flaps generated after the strand-transfer reaction" It is actually not known when these flaps are generated or (as alluded to above) whether they are generated at all in the first place.

Line 281, 285. "Interestingly" Please find some other way to arouse curiosity, rather than repetitive use of the same phrase as it is counterproductive.

Lines 290, 291. I think these lines try, in a somewhat convoluted way, to express the fact that the DNA binding is predominantly in-trans, so that NTD1 and NTD2 of one promoter are binding the transposon end that is processed by the catalytic subunit of another promoter, whose NTD1 and NTD2 are binding

the other transposon end. In fact, such in-trans binding is commonly observed in all DNA transposome structures that I'm aware of, so again, please use the standard terminology to describe the situation as it was developed in the field many years ago.

Line 296. "superfamilies 24" This reference is not appropriate here, as this Yuan&Wessler review was only about eukaryotic cut-and-paste DNA transposons, which V-K CAST is clearly not. In fact, the involvement of RNaseH like catalytic subunits with mobile genetic elements was discovered in the 1990s and there are many reviews relevant to prokaryotic systems.

Line 326. "appropriate distances" Is this what Fig4a is trying to show? If so, please put the figure reference here.

Line 353. I think the Pax6 connection could be moved up to around line 253 and settle the issue there, to avoid repetition of the same topic.

Line 368, 370 "end complex" Please use transposon end complex.

Line 441. "represent one of the most promising" Only from the point of view of the relatively small number of proteins needed. Considering the presence of co-integrates, one might argue that the system is actually much less promising than some of the others.

Line 466. "which previously shown to interact" As explained in the beginning, this is a significant outstanding issue in the field. Please expand the discussion of what the current structure might imply about this interaction and whether there are implications for how integration occurs at the target site.

Line 473. "could use" Are there any data suggesting that it does not use replicative transposition?

Line 478. "should accommodate TnsA" Why should it accommodate TnsA given that it doesn't encode it? In fact the similarity to the MuA STC complex, which also doesn't encode a nuclease to process the 5' end of the transposon, suggests that in fact V-K TnsB does not accommodate. Alternatively, are there any data suggesting that shTnsB interacts with any TnsA at all?

Line 518. "hairpins at the end of the transposon DNA 44" I believe that a V-K CAST TnsB making hairpins to facilitate double-strand breaks at the transposon ends is extremely unlikely. Reference 44 is not

based on data, but rather simply on a suggestion. As the authors are probably aware, all those cut-and-paste DNA transposases that generate hairpins to make DSBs contain an insertion domain in their RNaseH-like catalytic subunit that stabilizes the hairpin. shTnsB has no such insertion domain. Whether Dr. Zhang and colleagues are aware of this fact, I cannot comment.

Reviewer #2 (Remarks to the Author):

Tenjo-Castaño et al report a high resolution cryoEM structure of the TnsB-DNA complex of CAST V-K. The CASTs are of great interest and topicality, and many structural studies have been performed in the last couple of years (properly cited in this manuscript). This work only reports the TnsB-DNA complex structure, combined with some biochemical experiments, and does not present any information on the interaction between TnsB with other CAST components. As the authors themselves note towards the end of the manuscript, such interactions are of the most immediate interest for understanding the CAST mechanisms, and therefore, the novel insight from this work appears rather limited. Surely, there are interesting details reported on the conformation of the DDE catalytic sites in different TnsB protomers as well as useful information on the effects of mutation of DNA-binding residues. Nevertheless, it is not exactly clear from the current manuscript what is the novel insight compared to previous structural studies, in particular, those in their Ref. 19.

The work seems to be well done, from the purely experimental standpoint. However, the abstract of this manuscript is rather uninformative, whereas the Discussion section, while rather lengthy, is poorly focused. In the view of this reviewer, serious rewriting of at least those sections is required, focusing on the differences from previously described related transposases structures and mechanisms, and making the case for the novelty of the present findings

Reviewer #1

We thank the referee for his/her thorough evaluation and the comments about our manuscript. We also thank the referee for his/her suggestions, concerns, and corrections, which have improved our manuscript. We have addressed and incorporated all the points raised by the referee in the new version and hope he/she finds it suitable for publication.

Specific Comments:

Line 95. "Tn7-like transposons "parasite". I don't think this is really true, as who is parasitizing whom is a complex question and I'm not aware any data deciding the nature of this relationship. Tn7 is certainly not parasitizing anything for "mobilization" as Tn7 is perfectly capable of mobilizing itself. What can be said is that Tn7 has apparently evolved to employ a number of different site-specific targeting strategies and reliance on a crRNA is one of these.

We thank the referee for this comment. We have taken a literary license which leads to confusion. Therefore, we have eliminated this verb and substitute it by recruited in L96.

Line 121. "as it is unknown". In fact, the presence of co-integrates suggests that it is known, and very likely the 5' end is not processed by the transposase, and co-integrates are likely resolved by homologous recombination. Given that the authors have an in vivo assay in hand, they could have relatively easily asked the question whether the ratio of co-integrates changes if transposition was carried out in RecA- cells. One must point out that there are a number of replicative DNA transposon (IS5, IS6 families) that rely on resolution by host encoded systems; therefore, I do not see why something mysterious should be invoked unless data suggest that indeed something else beyond the obviously expected is going on.

The referee is right, and we do not have any data indicating any unknown differences. Therefore, we have rephrased L121-130 to avoid any mysterious mechanism including a mention to RecA mediated resolution. In the new text, we just pointed out the resemblance with Tn5053 and the possibility that an unknown host-encoded resolvase could be involved in providing the cointegrate resolution.

Line 154. Please rename left and right element as left and right ends, as is common in the literature. "Element" in the field typically refers to the mobile genetic element or MGE itself.

The name has been corrected according to the referee comment. See L161 in the revised version.

Line 157. "ladder of discrete bands, which increased" I do not know how a ladder increases; please rephrase.

The mistaken sentence has been corrected. See L164 in the revised version.

Line 160. "we tested whether shTnsB can bind independently". Please state what was the result.

We have stated the results of the experiment in L 167 in the new version “*Both complexes (shTnsB-RE and shTnsB-LE) were detected indicating that shTnsB the presence of both ends is not required for shTnsB to bind the repeats*”

Line 174. "TnsB and a RE and LE not bound to each other." Please rephrase, as I have no idea what this means.

We have rephrased this section to explain the experiment in which we tried to reconstitute the TnsB bound to the element ends. See L184-185 in the revised version.

Line 184. I do not find the description of the STC DNA construction easy to understand in the Methods. If I understand correctly, the STC DNA consists of six DNA strands, including two long transferred strands (ST). In these the transposon part is linked to the target part through a phosphodiester at the scissile phosphate. The non-transferred strands (NTS) are broken at the transposon/target junction with some bps missing to ensure base pairing, giving four other strands. This is the standard way to assemble STC DNA. Yet the terminology used in Methods speaks only about LE and RE, so it is not clear. I do not understand what "LE and RE ssDNA were mixed separately" means (line 611). Also, please use the TS and NTS terminology as this is common in the field. The most straight-forward might be to give the six DNA strands unique names, and describe from there.

We thank the referee for this comment. To address this point and clarify the reconstitution of the STC, we have explained the strategy in Results (L1985-203) and modified the Methods section “cryo-EM sample preparation” (L640-646), following the referee’s indications. Furthermore, we have included a new Supp. Figure, number 4, to fully describe the protocol of the assembly of the oligonucleotides building the STC DNA.

Furthermore, should change the "LTR" designation, as this is generally understood to mean something else altogether, denoting Long Terminal Repeats in retro-transposition and retrovirology.

We have decided to keep LTR for an easy comparison with Strecker et al Science 2019, which is the paper that first characterizes *S. hofmannii* type V-K CAST. In addition, we use the LTR as we want to differentiate them from the SR. To avoid any confusion, we have included a new panel with an alignment of the LTR and SRs. We think that this should be enough to avoid any confusion with the LTR term used in retro transposition and retrovirology

It is not clear how the target DNA sequence was chosen, so please describe this clearly. In particular, it is not clear what target site was chosen and why, or whether this particular site was in fact chosen with intent.

The target sequences for the reconstituted complex were chosen based on the natural sequences flanking the attachment site in the genome of *S. hofmannii* UTEX 2349, and the strategy was inspired in the method followed to obtain the P element transpososome structure (Ghanim et al., NSMB 2019). All this is described now in the revised version of the manuscript see the corresponding section in Methods and the L198-203 in results.

Really informative is Suppl. Fig6, where the STC DNA is finally shown (although - if I understand correctly - only the visible portion?). I think this figure should come much earlier.

We have modified this Supp Fig, which is sup fig 7 in the revised version. The new fig includes the bases contained in the oligonucleotide which are not visualized in the structure. We have tried to place this scheme earlier in the text, but it was not possible, we believe that the introduction of the new Supp fig 4 helps the reader to follow the narrative and the interaction scheme comes at an appropriate time as we need to include In between the cryo-EM validation and map quality figs.

I would like to know whether TS target DNA base pairing between the two TSs was intentionally designed as such, and if so why. What is the significance of this base pairing, and what is the significance if any of the apparently flipped out dA63 of the right NTS? Does this base-pairing and/or the flipped out base have any biological significance? Are these present at the biological target sites? If biologically significant, this could be a rather important finding in itself.

I note that the otherwise very useful Fig1d ignores this aspect of the STC design and structure, which is baffling.

We apologise for the scarce description of the STC design. We have expanded the description in the new version thoroughly. The sequences were designed from the natural site described by (Shamakov et al., Nature Rev. Microbiology 2017, Peters PNAS 2017, Strecker et al Science 2019), trying to obtain a meaningful physiological complex, this is now stated in Methods and further described in the new Supp. Fig 4.

Regarding the flipped out base mentioned by the referee. We apologise for the drawing in the former Supp Figure 6, as it might lead to confusion. There is no flipped out base, this base is an unpaired base. This uncoupling occurs because the natural duplicated site contains 5 bases, then this base located in the center of the duplicated site due cannot couple with its complementary base due to the STC DNA conformation. Therefore, we have modified that base in the new Supp Fig 7 scheme, and we have explained it in the corresponding figure legend to avoid any possible misunderstandings.

In general it appears that the authors have managed to generate complexes with one LE+target and one RE+target bound to it. The authors should explain why symmetrical complexes (LE/LE or RE/RE) bound were not observed and how they know for sure that in their observed complexes both LE+target and RE+target are indeed present.

The referee is concern about the possibility that we have a mixture of symmetrical and asymmetrical DNAs in our structure. We were also concern when we assembled the complex. However, if LE/LE or RE /RE complexes would be assembled with shTnsB, the central 5 nucleotides would need to be complementary, this is not the case in our target. No symmetry was applied in our cryo-EM processing and given the high resolution of the map (2.4 Å), and the fact that this region is the best local resolution (approx. 2.0 Å, see supp fig 6a), we are quite confident with this assignment. In addition, we have checked this possibility by building LE/LE and RE/RE sequences in this region and then proceeded with model building and refinement. The results of this *in silico* experiment show multiple deviation from the ideal geometry, and the sequence of the DNA shows problems in other regions of the structure. Furthermore, the possibility of a RE/RE or LE/LE DNA would imply that one of the central bases must be uncoupled (i.e., for the LE_Target, dT62/dC64 and for the RE_Target, dG61/dA63), and thereby the density of our unsymmetrized structure would present problems. This is not the case, and we need to build the LE/RE sequence to fulfil the density in that region (Fig.1 below). As it can be observed in the figure below the density displays a purine pyrimidine pair (see the different density of the two bases in the pair). In the case of a purine/purine or pyrimidine/pyrimidine pair, the density would look similar in both nucleotides, if not distorted by the base uncoupling. Therefore, we think that the best fitting corresponds to the LE/RE target that was built in our model.

Fig. 1. The figure displays the density map in the dC64-dG61 coupled pair.

Line 189. Fig 1a. This figure is not correct and misleading as it is drawn, as it implies that 5' cleavage somehow mysteriously occurred and carried out by shTnsB. There are no data for this. In fact, the likely scenario is that a Shapiro intermediate formed, leading to a co-integrate that is somehow resolved. I'm afraid that a more complicated figure needs be supplied that describes the likely strand cleavage and strand-transfer events.

We have modified Fig 1a according to the referee comments, please see the new version of Fig. 1a.

Line 220. "inferior ends" I do not know what inferior stands for.

The sentence referred to the branches of the X shaped structure. This has been corrected in L230-232 of the revised version of the manuscript "The structure of the shTnsB-STC complex resembles an elongated X with curved arms of different length. The longer arms correspond to the transposon ends while the short ones belong to the bent target DNA (Fig. 1d-e, 2a). In addition, we have modified figure 1d-e to avoid misunderstandings.

Line 222. "opposite branch" Please rephrase so it is compatible with the field's standard nomenclature.

The sentence has been corrected, see L238-240 of the revised version. We explain now that the location of the catalytic domain is *in trans*, as discussed in other papers about DNA transposition.

Line 245. "MyB/homeodomain like fold". Please give appropriate reference to this fold with a brief sentence as to what these proteins do.

The text has been corrected to explain the function of the MyB domain and a reference has been included (L263-264)

Line 246. "well-conserved". Please state that among what proteins R77 is conserved.

We have amended this sentence stating the proteins where the residue is conserved and citing Supp fig 1 (L265-267)

Line 253. "structure is not related" Please amend the negative statement with the results of the Dali search and state what this domain is actually related to.

We have amended the negative statement in the sentence including the results of the Dali analysis (L275-277).

Line 270. "5' flaps generated after the strand-transfer reaction" It is actually not know when these flaps are generated or (as alluded to above) whether they are generated at all in the first place.

The 5' flap terminology has been eliminated and we have termed them NTS (non-transferred strand) as advised by the referee. In addition, the referee is right, and we have removed the rest of the sentence alluding at the time and generation of this product, as there is a lack of knowledge regarding their generation, see L293.

Line 281, 285. "Interestingly" Please find some other way to arouse curiosity, rather than repetitive use of the same phrase as it is counterproductive.

We have substituted “Interestingly” for other synonyms to call the attention of the reader.

Lines 290, 291. I think these lines try, in a somewhat convoluted way, to express the fact that the DNA binding is predominantly in-trans, so that NTD1 and NTD2 of one promoter are binding the transposon end that is processed by the catalytic subunit of another promoter, whose NTD1 and NTD2 are binding the other transposon end. In fact, such in-trans binding is commonly observed in all DNA transposome structures that I'm aware of, so again, please use the standard terminology to describe the situation as it was developed in the field many years ago.

The referee is correct, we have simplified the text following the indications and using the standard terminology (L310-313).

Line 296. "superfamilies 24" This reference is not appropriate here, as this Yuan&Wessler review was only about eukaryotic cut-and-paste DNA transposons, which V-K CAST is clearly not. In fact, the involvement of RNaseH like catalytic subunits with mobile genetic elements was discovered in the 1990s and there are many reviews relevant to prokaryotic systems.

Thanks for the observation. We have provided a more appropriate references in this sentence (L317)

Line 326. "appropriate distances" Is this what Fig4a is trying to show? If so, please put the figure reference here.

Yes, this is now corrected in L347 in the new version.

Line 353. I think the Pax6 connection could be moved up to around line 253 and settle the issue there, to avoid repetition of the same topic.

We have corrected this point in a different way, as we think that a detailed explanation here belongs better to the discussion section. So, in the previous point related to these similitudes we explained more generally the Pax domain.

Line 368, 370 "end complex" Please use transposon end complex.

This terminology has been corrected in L388, 390, 456, 491 and 778

Line 441. "represent one of the most promising" Only from the point of view of the relatively small number of proteins needed. Considering the presence of co-integrates, one might argue that the system is actually much less promising than some of the others.

The sentence has been modified to incorporate the co-integrates problem as suggested by the referee, see L451-453.

Line 466. "which previously shown to interact" As explained in the beginning, this is a significant outstanding issue in the field. Please expand the discussion of what the current structure might imply about this interaction and whether there are implications for how integration occurs at the target site.

The sentence has been eliminated and the whole discussion has been rewritten. The discussion about the TnsC and TnsB interaction has been expanded including a new hypothesis about integration L496-526 and Fig. 7d,

Line 473. "could use" Are there any data suggesting that it does not use replicative transposition?

Corrected, both "could" have been deleted L471

Line 478. "should accommodate TnsA" Why should it accommodate TnsA given that it doesn't encode it? In fact the similarity to the MuA STC complex, which also doesn't encode a nuclease to process the 5' end of the transposon, suggests that in fact V-K TnsB does not accommodate. Alternatively, are there any data suggesting that shTnsB interacts with any TnsA at all?

There is a misunderstanding in this criticism. We are not referring to shTnsB but to Tn7-ecTnsB, which does code TnsA. "Therefore, the structural similarities of the overall STC complex between shTnsB and MuA would be supported by these mechanistic similitudes. Yet, we will expect that the Tn7-STC complex should accommodate TnsA. From our structure it is not clear how the 5'-ends could be processed in the Tn7 system"

There are several papers indicating the interaction of TnsA and TnsB of in the Tn7 system, for example (Choi et al., PNAS 2013). We have rephrased this sentence to avoid confusions (L475-478).

Line 518. "hairpins at the end of the transposon DNA 44" I believe that a V-K CAST TnsB making hairpins to facilitate double-strand breaks at the transposon ends is extremely unlikely. Reference 44 is not based on data, but rather simply on a suggestion. As the authors are probably aware, all those cut-and-paste DNA transposases that generate hairpins to make DSBs contain an insertion domain in their RNaseH-like catalytic subunit that stabilizes the hairpin. shTnsB has no such insertion domain. Whether Dr. Zhang and colleagues are aware of this fact, I cannot comment.

We agree with the referee. In fact, that's what we meant in this paragraph. The suggestion by Zhang and colleagues is not supported by the evidence. As pointed out by the referee. We have modified the text to make this argument clearer (L532-535).

Reviewer #2

We thank the referee for his/her evaluation and the comments about our manuscript. We also thank the referee for praising the quality of our experimental approach. We have addressed and incorporated the points raised by the referee in the new version and hope he/she finds it suitable for publication.

Tenjo-Castaño et al report a high resolution cryoEM structure of the TnsB-DNA complex of CAST V-K. The CASTs are of great interest and topicality, and many structural studies have been performed in the last couple of years (properly cited in this manuscript). This work only reports the TnsB-DNA complex structure, combined with some biochemical experiments, and does not present any information on the interaction between TnsB with other CAST components. As the authors themselves note towards the end of the manuscript, such interactions are of the most immediate interest for understanding the CAST mechanisms, and therefore, the novel insight from this work appears rather limited. Surely, there are interesting details reported on the conformation of the DDE catalytic sites in different TnsB protomers as well as useful information on the effects of mutation of DNA-binding residues. Nevertheless, it is not exactly clear from the current manuscript what is the novel insight compared to previous structural studies, in particular, those in their Ref. 19. The work seems to be well done, from the purely experimental standpoint. However, the abstract of this manuscript is rather uninformative, whereas the Discussion section, while rather lengthy, is poorly focused. In the view of this reviewer, serious rewriting of at least those sections is required, focusing on the differences from previously described related transposases structures and mechanisms, and making the case for the novelty of the present findings

The referee acknowledges the quality of our experimental data and the approach. He/she is not concern with that part of our manuscript. However, he/she feels that we have not discussed well our findings and we must make a better case for our new findings in the manuscript.

Therefore, following the indications of Reviewer 2 and also addressing one of the points of Reviewer 1, who asked us to discuss more thoroughly the role of shTnsB in the type V-K CAST mechanism, we have rewritten the abstract and the complete discussion commenting the two models available of how RNA-guided transposition is on the light of our data and we have also proposed a new variant to those models based in the interaction between shTnsB and shTnsC (see the discussion section and new Fig 7b). Regarding the inclusion of a comparison with other transposases in the Discussion section, as suggested by the referee, we think that the paper contains already 3 main figures (Fig.4-6) and Supplementary Fig 1, where we make a comparison between the different catalytic domains and the DNA binding with the close homologues. Thus, we feel that this comparison is well-covered in results.

REVIEWERS' COMMENTS

Reviewer #1 (Remarks to the Author):

This is a substantially improved and clarified version of the manuscript.

I have only a few minor comments.

Line 97. "These RNP complexes" I think here the context is the entire transposon, so it is not really just an RNP complex.

Line 133. Please insert 3' between free and hydroxyl.

Line 135. I suggest replacing "attachment" with target.

Line 175. How about: "...shift and cannot be excluded as a possibility."

Line 235. "OD domain" I think it might be better to define the domain abbreviations in the main text, rather than in the figure legend.

Line 254. "This helix" Which helix is that?

Line 325. I think only panel e of suppl fig8 might be relevant here.

Line 409. Can we have an arrow or some such pointing to that beta strand?

Line 499. I do not not think Suppl fig3 is the correct one here.

Line 534. Please write as "of the insertion domain that stabilizes"

Figure 2a still has "left element" and "right element" Please change to Left End and Right End.

Reviewer #1

We thank the referee for his/her thorough evaluation and the comments about our manuscript. We also thank the referee for his/her suggestions, concerns, and corrections, which have improved our manuscript. We have incorporated all the minor points raised by the referee in the new version

Line 97. "These RNP complexes" I think here the context is the entire transposon, so it is not really just an RNP complex.

Corrected

Line 133. Please insert 3' between free and hydroxyl.

Corrected

Line 135. I suggest replacing "attachment" with target.

Corrected

Line 175. How about: "...shift and cannot be excluded as a possibility."

Corrected

Line 235. "OD domain" I think it might be better to define the domain abbreviations in the main text, rather than in the figure legend.

Corrected

Line 254. "This helix" Which helix is that?

Corrected

Line 325. I think only panel e of suppl fig8 might be relevant here.

Corrected

Line 409. Can we have an arrow or some such pointing to that beta strand?

Corrected

Line 499. I do not not think Suppl fig3 is the correct one here.

Corrected

Line 534. Please write as "of the insertion domain that stabilizes"

Corrected

Figure 2a still has "left element" and "right element" Please change to Left End and Right End.

Corrected